# Interspecies conservation of organisation and function between nonhomologous regional centromeres

Pin Tong[1,6], Alison L. Pidoux[1,6], Nicholas R.T. Toda[1,3], Ryan Ard [1,4], Harald Berger[1,5], Manu Shukla[1], Jesus Torres-Garcia [1], Carolin A. Müller [2], Conrad A. Nieduszynski [2] & Robin C. Allshire [1]

Despite the conserved essential function of centromeres, centromeric DNA itself is not conserved. The histone-H3 variant, CENP-A, is the epigenetic mark that specifies centromere identity. Paradoxically, CENP-A normally assembles on particular sequences at specific genomic locations. To gain insight into the specification of complex centromeres, here we take an evolutionary approach, fully assembling genomes and centromeres of related fission yeasts. Centromere domain organization, but not sequence, is conserved between *Schizosaccharomyces pombe*, *S. octosporus* and *S. cryophilus* with a central CENP-A$^{Cnp1}$ domain flanked by heterochromatic outer-repeat regions. Conserved syntenic clusters of tRNA genes and 5S rRNA genes occur across the centromeres of *S. octosporus* and *S. cryophilus*, suggesting conserved function. Interestingly, nonhomologous centromere central-core sequences from *S. octosporus* and *S. cryophilus* are recognized in *S. pombe*, resulting in cross-species establishment of CENP-A$^{Cnp1}$ chromatin and functional kinetochores. Therefore, despite the lack of sequence conservation, *Schizosaccharomyces* centromere DNA possesses intrinsic conserved properties that promote assembly of CENP-A chromatin.

---

[1] Wellcome Centre for Cell Biology and Institute of Cell Biology, School of Biological Sciences, The University of Edinburgh, Mayfield Road, Edinburgh EH9 3BF, UK. [2] Sir William Dunn School of Pathology, University of Oxford, South Parks Road, Oxford OX1 3RE, UK. [3]Present address: UPMC CNRS, Roscoff Marine Station, Place Georges Teissier, 29680 Roscoff, France. [4]Present address: Copenhagen Plant Science Centre, University of Copenhagen, Bülowsvej 34, 1870 Frederiksberg C, Denmark. [5]Present address: Symbiocyte, Universität für Bodenkultur Wien, University of Natural Resources and Life Sciences, 1180 Vienna, Austria. [6]These authors contributed equally: Pin Tong, Alison L. Pidoux. Correspondence and requests for materials should be addressed to A.L.P. (email: alison.pidoux@ed.ac.uk) or to R.C.A. (email: robin.allshire@ed.ac.uk)

Centromeres are the chromosomal regions upon which kinetochores assemble to mediate accurate chromosome segregation. Evidence suggests that both genetic and epigenetic influences define centromere identity[1–9]. Neocentromere formation at new locations lacking homology to centromeres[10] and the inactivation of one centromere of a dicentric chromosome despite it retaining centromeric sequences[11] indicate that centromere sequences are neither necessary nor sufficient for centromere assembly[1,7,9]. CENP-A is found at all active centromeres and is the epigenetic mark that specifies centromere identity[1,7,9]. Artificial tethering of CENP-A or CENP-A loading factors at non-centromeric locations on metazoan chromosomes is sufficient to trigger kinetochore assembly[5,12]. Thus, it is specialized chromatin rather than primary sequences of centromeric DNA that determines where kinetochores and hence functional centromeres are assembled. On the contrary, however, CENP-A is generally found on particular sequences in any given organism[1,2,4] and naked repetitive centromere DNA such as alpha-satellite DNA can provide a substrate for the de novo assembly of functional centromeres when introduced into human cells[1–4]. These observations suggest that, despite the lack of conservation between species, centromere sequences possess properties that make them attractive for assembly of CENP-A chromatin.

*Schizosaccharomyces pombe*, a paradigm for dissecting complex regional centromere function, has demarcated centromeres (35–110 kb) with a central domain assembled in CENP-A$^{Cnp1}$ chromatin, flanked by outer-repeat elements assembled in RNA interference-dependent heterochromatin, in which histone-H3 is methylated on lysine-9 (H3K9)[13–16]. Heterochromatin is required for establishment but not maintenance of CENP-A$^{Cnp1}$ chromatin[6,17]. We have proposed that it is not the sequence per se of *S. pombe* central-core that is key in its ability to establish CENP-A chromatin but the properties programmed by it[18,19].

To investigate whether these properties are conserved, here we completely assemble the sequence across centromeres of other *Schizosaccharomyces* species and test their cross-species functionality. We show that although *Schizosaccharomyces* centromeres are not conserved in sequence, those of *Schizosaccharomyces octosporus* and *Schizosaccharomyces cryophilus* share with *S. pombe* a conserved organization of a central domain assembled in CENP-A$^{Cnp1}$ chromatin, flanked by outer repeats assembled in heterochromatin. Syntenic clusters of tRNA and 5S-rRNA genes are present across *S. octosporus* and *S. cryophilus* centromeres, further emphasizing their conserved organization. By introducing minichromosomes bearing central domain sequences from *S. octosporus* and *S. cryophilus* into *S. pombe*, we demonstrate that these nonhomologous centromere sequences can be recognized between divergent species, allowing the establishment of CENP-A$^{Cnp1}$ chromatin and functional centromeres. These observations indicate that centromere DNA possesses conserved properties that promote the establishment of CENP-A chromatin.

## Results

**Conserved organization of fission yeast centromeres.** Long-read (PacBio) sequencing permitted complete assembly of the genomes across centromeres of *S. octosporus* (11.9 Mb) and *S. cryophilus* (12.0 Mb), extending genome sequences[20] to telomeric or subtelomeric repeats, or rDNA arrays (Supplementary Figs. 1–3, Supplementary Data 1, 2). Consistent with their closer evolutionary relationship[20–22], *S. octosporus* and *S. cryophilus* (32 My separation, compared with 119 My separation from *S. pombe*) exhibit greatest synteny (Fig. 1a), in agreement with a recent report in which joining of *S. cryophilus* supercontigs[20] into

chromosome arm-sized assemblies and comparative analysis identified translocations and inversion events that occurred during divergence of fission yeast species[22]. Synteny is preserved adjacent to centromeres (Fig. 1b). Circos plots indicate a chromosome arm translocation occurred within two ancestral centromeres to generate *S. cryophilus cen2* (*S.cry-cen2*) and *S.cry-cen3* relative to *S. octosporus* and *S. pombe* (Fig. 1b). Despite centromere-adjacent synteny, *Schizosaccharomyces* centromeres lack detectable sequence homology (see below). All centromeres contain a central domain: central core (*cnt*) surrounded by inverted repeat (*imr*) elements unique to each centromere (Fig. 2, Supplementary Fig. 4, Supplementary Tables 1,2, Supplementary Data 3,4). CENP-A$^{Cnp1}$ localizes to fission yeast centromeres (Fig. 2a) and chromatin immunoprecipitation sequencing (ChIP-Seq) indicates that central domains are assembled in CENP-A$^{Cnp1}$ chromatin, flanked by various outer-repeat elements assembled in H3K9me2 heterochromatin (Fig. 2b, c). Despite the lack of sequence conservation, *S. octosporus* and *S. cryophilus* centromere organization is strongly conserved with that of *S. pombe*, having CENP-A$^{Cnp1}$-assembled central domains separated by clusters of tRNA genes from outer repeats assembled in heterochromatin[13,14] (Supplementary Fig. 5, Supplementary Table 3, Supplementary Data 5). In contrast, our analyses of partially assembled, transposon-rich centromeres of *Schizosaccharomyces japonicus* reveals the presence of heterochromatin on all classes of retrotransposons and CENP-A$^{Cnp1}$ on only two classes (Tj6 and Tj7; Supplementary Fig. 6, Supplementary Table 4)[20].

**Syntenic clusters of tRNA genes at centromeres.** Numerous 5S rRNA genes (5S rDNAs) are located in the heterochromatic outer repeats of *S. octosporus* and *S. cryophilus* centromeres (but not *S. pombe*) (Fig. 1a, Supplementary Data 6, 7). Almost all (25/26; 20/20) are within Five-S-Associated Repeats (FSARs; 0.6–4.2 kb) (Fig. 3a), encompassing ~35% of outer-repeat regions. FSARs exhibit 90% intra-class homology (Supplementary Table 5) but no interspecies homology. The three types of FSAR repeats almost always occur together, in the same order and orientation, but vary in copy number: *S. octosporus*: (oFSAR-1)$_1$(oFSAR-2)$_{1–9}$(oFSAR-3)$_1$; *S. cryophilus*: (cFSAR-1)$_{1–3}$(cFSAR-2)$_{1–2}$(cFSAR-3)$_1$. Both sides of *S. octosporus* and *S. cryophilus* centromeres contain at least one FSAR-1-2-3 array, except the right side of *S.cry-cen2* with two lone cFSAR-3 elements (Fig. 3a, Supplementary Fig. 4). *S. cryophilus* cFSAR-2 and cFSAR-3 repeats share ~400 bp homology (88% identity), constituting *hsp16* heat-shock protein open reading frames (ORFs) (Fig. 3a, b, Supplementary Data 8) that are intact, implying functionality, selection and expression in some situations. Phylogenetic gene trees indicate that cFSAR-3-*hsp16* genes are more closely related with each other than with those in subtelomeric regions or cFSAR-2s (Fig. 3b), consistent with repeat homogenization[23–25]. cFSAR-1s contain an eroded ORF with homology to a small hypothetical protein and *S. octosporus* oFSAR-2s contain a region of homology with a family of membrane proteins (Fig. 3a). The functions of centromere-associated *hsp16* genes and other ORF-homologous regions remain to be explored.

*S. cryophilus* heterochromatic outer repeats contain additional repetitive elements, including a 6.2 kb element (cTAR-14) with homology to the retrotransposon *Tcry1* and transposon remnants at the mating-type locus[20] (Figs. 1a, 2b, Supplementary Fig. 4, Supplementary Tables 1,6 and Supplementary Data 3). *Tcry1* is located in the chrIII-R subtelomeric region (Supplementary Figs. 3, 4 and Supplementary Data 1). Although no retrotransposons have been identified in *S. octosporus*, remnants are present in the mating-type locus and oTAR-14ex in *S.oct-cen3*

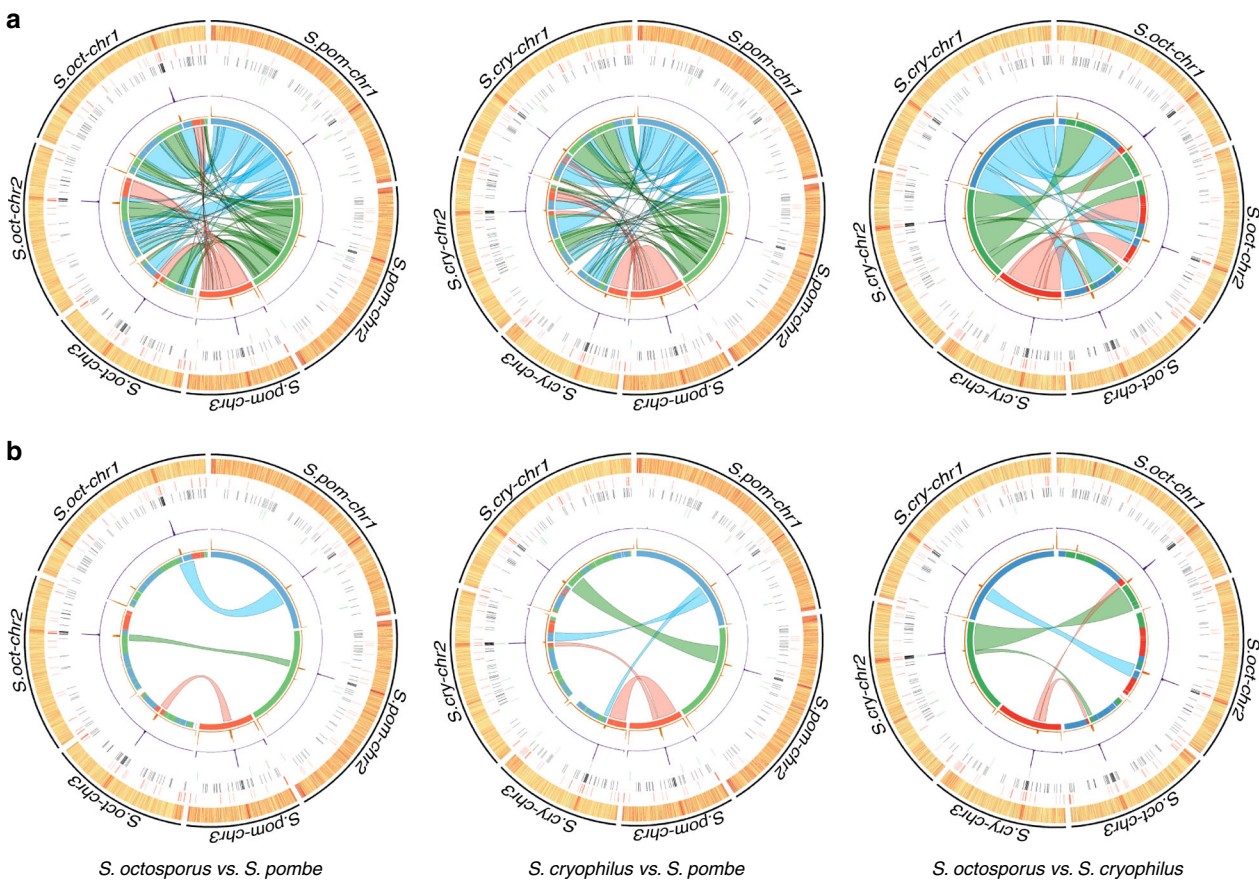

**Fig. 1** Genome organization and synteny in *Schizosaccharomyces*. **a** Circos plots depicting pairwise *S. pombe*, *S. octosporus* and *S. cryophilus* genome synteny. Rings from outside to inside represent the following: chromosomes; GC content (high: red, low: yellow); 5S rDNAs (red); tRNA genes (black); LTRs (green); CENP-A$^{Cnp1}$ ChIP-seq (purple); H3K9me2 ChIP-seq (orange); innermost ring and coloured connectors indicate regions of synteny between species. *S. pombe* chromosomes are indicated by blue (*S.pom-chr1*), green (*S.pom-chr2*), red (*S.pom-chr3*) in the left and right panels, and regions of synteny on *S. octosporus* and *S. cryophilus* chromosomes, respectively, are indicated in corresponding colours. A similar designation is used for *S. octosporus* chromosomes in the middle panel. **b** Circos plot isolating regions adjacent to centromeres highlighting preserved synteny and an intra-centromeric chromosome arm swap involving *S. cryophilus cen2* and *cen3* relative to *S. pombe* and *S. octosporus*. Source data available: GEO: GSE112454

outer repeats (Fig. 2c, Supplementary Figs. 1, 4, Supplementary Tables 2, 7 and Supplementary Data 4). Hence, transposon remnants, FSARs and other repeats are assembled in heterochromatin at *S. octosporus* and *S. cryophilus* centromeres, and potentially mediate heterochromatin nucleation.

tRNA gene clusters occur at transitions between CENP-A and heterochromatin domains in two of three centromeres in *S. octosporus* (*S.oct-cen2*, *S.oct-cen3*) and *S. cryophilus* (*S.cry-cen1*, *S.cry-cen2*), and are associated with low levels of both H3K9me2 and CENP-A$^{Cnp1}$ (Fig. 2b, c), suggesting that they may act as boundaries, as in *S. pombe*[26–28]. No tRNA genes demarcate the CENP-A/heterochromatin transition at *S.cry-cen3*. Instead, this transition coincides precisely with 270 bp LTRs (Fig. 2b, Supplementary Tables 1, 6 and Supplementary Data 3), which may also act as boundaries[29–31]. Similar to tRNA genes, LTRs have been shown to be regions of low nucleosome occupancy, which may counter spreading of heterochromatin[31,32]. The transition between CENP-A$^{Cnp1}$ and heterochromatin is poorly demarcated at *S.oct-cen1* compared with other centromeres. This region lacks tRNA genes and, as only retrotransposon remnants are detectable in *S. octosporus*, the sequence of putative LTRs is unknown. It is possible that the long inverted *imr* repeats comprise a gradual transition zone at this centromere. tRNA gene clusters also occur near the extremities of all centromeres in both species, separating heterochromatin from adjacent euchromatin.

tRNA genes and LTRs are thus likely to act as chromatin boundaries at fission yeast centromeres.

A high proportion (~32%) of tRNA genes in *S. pombe*, *S. octosporus* and *S. cryophilus* genomes are located within centromere regions[33] (Figs. 1a, 3c, Supplementary Table 8 and Supplementary Data 9, 10). Centromeric tRNA genes are intact and are conserved in sequence with their genome-wide counterparts, indicating that they are functional genes. Two major, conserved tRNA gene clusters reside exclusively within *S. octosporus* and *S. cryophilus* centromeres (*p*-value < 0.00001; *q*-value < 0.05) (Fig. 3c, d). Cluster 1 comprises several subclusters of 2–3 tRNA genes in various combinations of up to 8 tRNA genes, whereas Cluster2 contains up to 5 tRNA genes (Fig. 3d); 17 different tRNA genes (14 amino acids) are represented, none of which are unique to centromeres (Fig. 3c). Intriguingly, the order and orientation of tRNA genes within clusters is conserved between species, but intervening sequence is not (Fig. 3d, e). Strikingly, as well as local tRNA gene cluster conservation, inspection of centromere maps reveals synteny of tRNA genes and clusters across large portions of *S. octosporus* and *S. cryophilus* centromeres. For example, the tRNA gene order AIR-RKL-E-T-T-L-DVAIR-RKLEF-A-DV (single-letter code) is observed at *S.oct-cen1* and *S.cry-cen3* (Supplementary Fig. 7). This synteny, together with both possessing small central cores and long *imrs*, suggests that these two centromeres are ancestrally

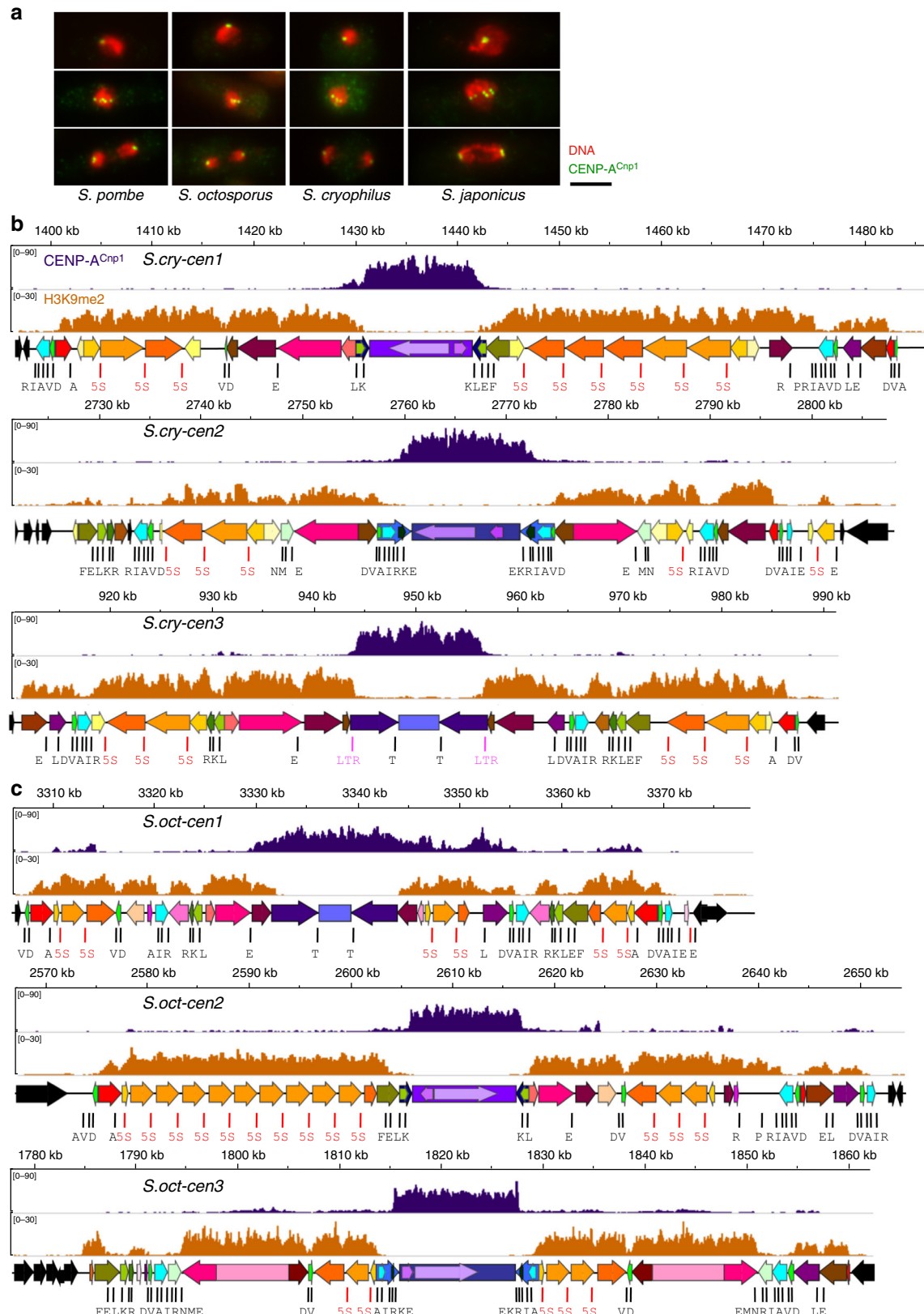

related (Fig. 3f). Similarly, at *S.oct-cen3* and *S.cry-cen2*, tRNA genes occur in the order NME-DV-AIRKE-EKRIA-VD-EMN-RIAVD, and at *S.oct-cen2* and *S.cry-cen1* the same tRNA genes are present in the *imr* repeats and beyond (FELK-KL-E-DV). Central cores have similar sizes and structures in the two species,

each containing long (oCNT-L(6.4 kb); cCNT-L(6.0 kb)) and short (oCNT-S(1.2 kb); cCNT-S(1.3 kb)) species-specific repeats (Fig. 3f, Supplementary Tables 1, 2, 9 and Supplementary Data 3, 4). CNT repeats are arranged head-to-tail at one centromere and head-to-head at the other centromere in each species.

**Fig. 2** Domain organization of *Schizosaccharomyces* centromeres. **a** Immunostaining of centromeres in indicated *Schizosaccharomyces* species with anti-CENP-A[Cnp1] antibody (green) and DNA staining (DAPI; red). Scale bar, 5 μm. **b** *S. cryophilus* centromere organization indicating DNA repeat elements. ChIP-seq profiles for CENP-A[Cnp1] (purple) and H3K9me2-heterochromatin (orange) are shown above each centromere. Positions of tRNA genes (single-letter code of cognate amino acid; black), 5S rDNAs (red) and solo LTRs are indicated (pink). Central cores (cnt—purples) innermost repeats (imr—blue shades). 5S-associated repeats (cFSARs—orange shades); tRNA gene-associated repeats (TARs) containing clusters of tRNA genes (green shades); heterochromatic repeats (cHR) and TARs associated with single tRNA genes (various colours: brown/pink/red). cTAR-14s, containing retrotransposon remnants (deep pink). For details, including individual repeat annotation, see Supplementary Fig. 4 and Supplementary Tables 3,4. **c** *S. octosporus* centromere organization indicating DNA repeat elements. Labelling and shading as in **b**. Only oTAR-14ex (pale pink part) contain retrotransposon remnants. Colouring is indicative of homology within each species but only of possible repeat equivalence (not homology) between species; see Supplementary Table 5,6,19. Source data available: GEO: GSE112454 and in Source Data file

Together, these similarities suggest ancestral relationships between *S. oct-cen2* and *S. cry-cen1*, *So-cen3* and *Scry-cen2*. Further, in places where synteny appears to break down, patterns of tRNA gene clusters suggest specific centromeric rearrangements occurred between the species. For instance, tRNA gene clusters at the edges of *S. cry-cen2R* and *S. cry-cen3L* are consistent with an inter-centromere arm translocation relative to *S. oct-cen1R* and *S. oct-cen2R*, indicated by gene synteny maps (Figs. 1b, 4a and Supplementary Fig. 7).

**Fission yeast centromeres show interspecies functionality.** No central-core sequence homology was revealed between species using BLASTN. To identify potential underlying centromere sequence features, k-mer frequencies (5-mers), normalized for centromeric AT-bias, were subjected to principal component analysis (PCA). CENP-A[Cnp1]-associated regions of *S. pombe*, *S. octosporus* and *S. cryophilus* genomes all group together, distinct from the majority of non-centromere sequences (*p*-value, $9.3 \times 10^{-7}$) (Fig. 4b, c). Interestingly, *S. pombe* neocentromere-forming regions[34] also cluster separately from other genomic regions, sharing sequence features with centromeres. Surprisingly, taking GC content into account, the *S. japonicus* genome as a whole shows no significant difference in 5-mer frequency compared with the other three fission yeast genomes. In contrast, *S. japonicus* CENP-A[Cnp1]-associated 5-mer frequencies show significant differences from its own wider genome sequence and from centromere sequences of the other fission yeast species (Supplementary Fig. 6).

K-mer analysis and conserved centromeric organization prompted us to investigate cross-species functionality of protein and DNA components of *Schizosaccharomyces* centromeres. green fluorescent protein (GFP)-tagged CENP-A[Cnp1] protein from each species localized to *S. pombe* centromeres and complemented the *cnp1-1* mutant[35] (Fig. 5a–c), indicating that heterologous CENP-A proteins assemble and function at *S. pombe* centromeres, despite normally assembling on non-homologous sequences in their respective organisms.

Introduction of *S. pombe* central-core (*S.pom-cnt*) DNA on minichromosomes into *S. pombe* results in the establishment and maintenance of CENP-A[Cnp1] chromatin if *S.pom-cnt* is adjacent to heterochromatin, or if CENP-A is overexpressed[6,17,18,36]. *S.oct-cnt* regions (3.2–10 kb) or *S.pom-cnt2* (positive control) were placed adjacent to *S. pombe* outer-repeat DNA in minichromosome constructs (Fig. 6a), which were transformed into *S. pombe* cells overexpressing *S. pombe* GFP-CENP-A[Cnp1] (hi-CENP-A[Cnp1])[18]. Acquisition of centromere function is indicated by minichromosome retention on non-selective indicator plates (white/pale pink colonies) and by the appearance of sectored colonies (Fig. 6b, c). The pHET-*S.pom-cnt2* minichromosome containing *S.pom-cnt2* established centromere function at high frequency immediately upon transformation in hi-CENP-A[Cnp1] cells (Table 1). Centromere function was also established on *S.oct-cnt*-containing minichromosomes in hi-CENP-A[Cnp1]

(Fig. 6b, c and Table 1). CENP-A[Cnp1] ChIP-quantitative PCR (ChIP-qPCR) indicated that, for minichromosomes with established centromere function, CENP-A[Cnp1] chromatin was assembled on nonhomologous *S.oct-cnt* DNA, to levels similar to those at endogenous *S. pombe* centromeres and to *S.pom-cnt2* on a minichromosome (Fig. 6d). Minichromosomes containing *S.oct-cnt* provided efficient segregation function (Table 1), no longer requiring CENP-A[Cnp1] overexpression to maintain that function once established (Fig. 6e), consistent with the self-propagating ability of CENP-A chromatin[5,18]. Minichromosomes containing *S. cryophilus* central-core regions (*S.cry-cnt*) were also able to establish functional centromeres and segregation function in *S. pombe*. These *S.cry-cnt*-bearing minichromosomes assembled CENP-A[Cnp1] chromatin to high levels, similar to those at endogenous *S. pombe* centromeres (Supplementary Fig. 8). Centromere function was not due to minichromosomes gaining portions of *S. pombe* central-core DNA (Supplementary Fig. 9). A similar minichromosome bearing a region (retrotransposon Tj7) highly enriched for CENP-A[Cnp1] in *S. japonicus* did not convincingly form functional centromeres when introduced into *S. pombe* or assemble CENP-A[Cnp1] chromatin to an appreciable extent (Supplementary Fig. 6). Thus, *S. pombe*, *S. octosporus* and *S. cryophilus* centromeres share a similar organization, underlying sequence features and cross-species establishment of CENP-A[Cnp1] chromatin, whereas putative *S. japonicus* centromeres appear not to share these attributes. Our analyses indicate that *S.oct-cnt* and *S.cry-cnt* DNAs are competent to establish CENP-A chromatin and centromere function in *S. pombe* when CENP-A[Cnp1] is overexpressed, suggesting that *S. octosporus* and *S. cryophilus* central-core DNA have intrinsic properties that promote the establishment of CENP-A chromatin despite lacking sequence homology.

## Discussion
Our analyses indicate that the centromeres of *S. pombe*, *S. octosporus* and *S. cryophilus* share a conserved organization of a CENP-A[Cnp1]-assembled central-core flanked by outer repeats assembled in H3K9me heterochromatin. Despite this conservation of organization, centromere sequence is not conserved, although underlying sequence features are detectable by PCA of 5-mer frequencies. The cross-species functionality of *S.oct-cnt* and *S.cry-cnt* central-core DNA in *S. pombe* suggests that the central-core regions of these three species are favoured substrates, sharing intrinsic properties that promote the establishment of CENP-A[Cnp1] chromatin, properties that *S.jap-Tj7* may lack. Although the nature of putative conserved CENP-A-promoting properties is unknown, recent studies have revealed distinctive characteristics of centromeric DNA. *S. pombe* central-core DNA has the innate property of driving high rates of histone-H3 nucleosome turnover, causing low nucleosome occupancy[19] and may programme pervasive low-quality RNAPII transcription to promote assembly of CENP-A chromatin[18]. These and other properties, such as non-B form DNA[37], may contribute to an

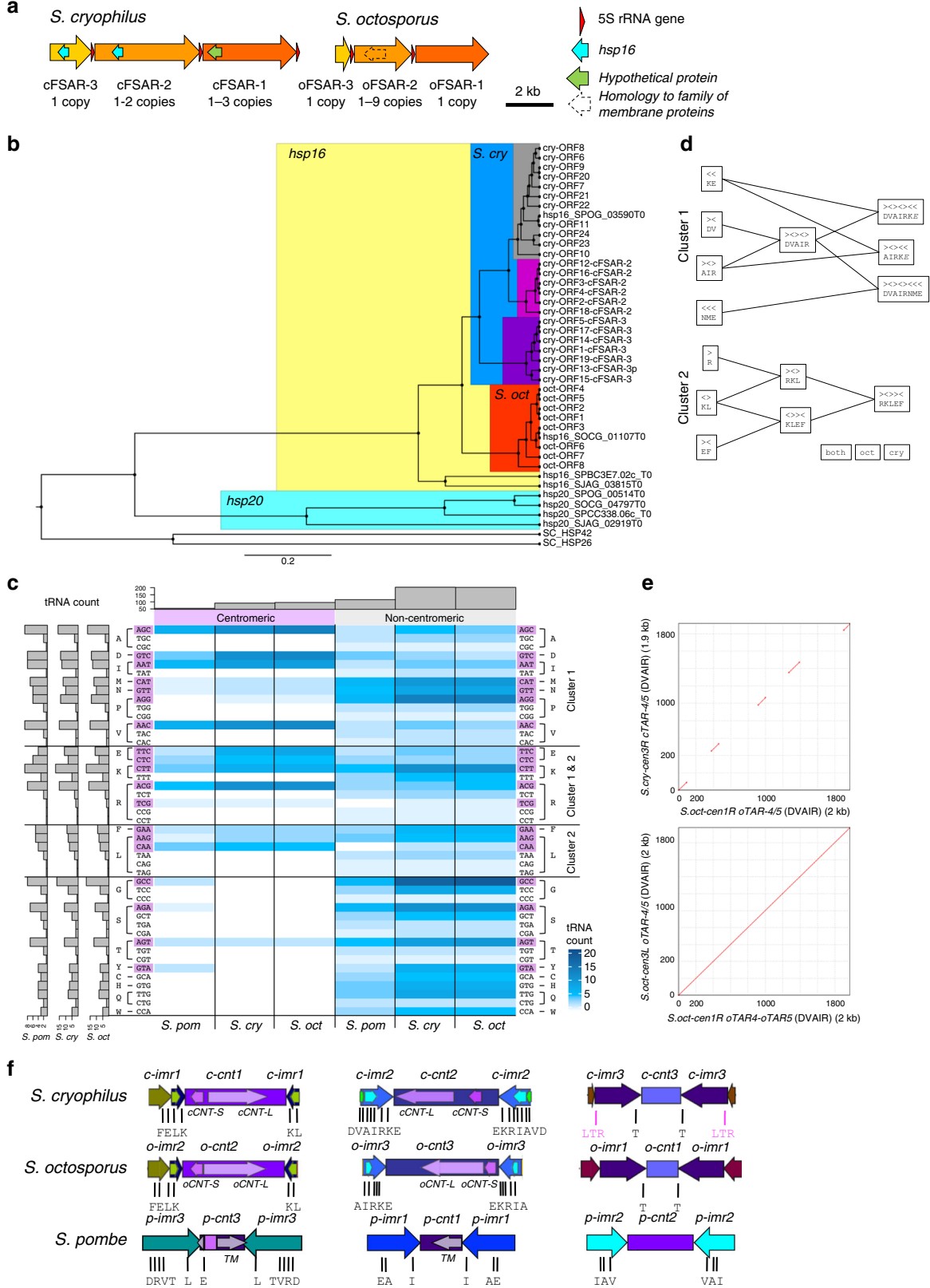

intrinsic CENP-A deposition programme conserved between *Schizosaccharomyces* centromeric DNA.

Based on conserved features, ancestral *Schizosaccharomyces* centromeres may have consisted of a CENP-A$^{Cnp1}$-assembled central-core surrounded by tRNA gene clusters and 5S rDNAs. We surmise that RNAPIII promoters perhaps provided targets for transposon integration[38], followed by heterochromatin formation to silence retrotransposons and preserve genome integrity[39,40]. The ability of heterochromatin to recruit cohesin[41,42], benefitting chromosome segregation, selected for heterochromatin maintenance[43,44], rather than selecting for underlying sequence, which evolved by repeat expansion and continuous

**Fig. 3** *S. cryophilus* and *S. octosporus* contain conserved clusters of tRNA genes and similar nonhomologous repeat elements. **a** Schematic of *S. cryophilus* and *S. octosporus* FSAR repeats, indicating positions of 5S rDNAs, *hsp16* genes and other ORFs. Copy number of each FSAR within centromeric arrays is indicated. **b** Phylogenetic relationship of *S. cryophilus* centromeric *hsp16* genes with genomic *hsp16* and *hsp20* genes of *S. cryophilus*, *S. octosporus*, *S. pombe* and *S. japonicas*. **c** Heat map of tRNA gene frequency at centromeric and non-centromeric sites (blue shades) for *S. pombe*, *S. cryophilus* and *S. octosporus*. Anticodons and cognate amino acids indicated right (purple: present at centromeres). Clusters containing these tRNA genes indicated. Histogram (top): total tRNA gene frequencies in centromeres and non-centromeric sites of indicated species. Histogram (left): tRNA gene frequencies in each species. **d** Depiction of centromeric tRNA gene clusters and subclusters. Combinations of 2 or 3 tRNA genes subclusters present in both species (purple) or specific to *S. octosporus* (red) or *S. cryophilus* (blue) are indicated (single-letter code of cognate amino acid; arrows indicate plus or minus strand). **e** Top: Dot-plot alignment (MEGABLAST) showing synteny between oTAR-4/oTAR-5 (DVAIR-Cluster 1) from *S.oct-cen1R* (chr1:3355194-3357165) with oTAR-4/oTAR-5 (DVAIR-Cluster 1) from *S.cry-cen3R* (chr3:964707-966623). Bottom: Dot-plot of oTAR-4/oTAR-5 (DVAIR-Cluster 1) from *S.oct-cen1R* (chr1:3355194-3357165) and oTAR-4/oTAR-5 (DVAIR-Cluster 1) from *S.oct-cen3L* (chr3:1791072-1793051). **f** Schematic of central domain similarity between species. Central cores (purple shades), *imr* (blues), TARs containing tRNA gene clusters (greens). Long (*CNT-L*) and short (*CNT-S*) central-core repeats are indicated. tRNA genes indicated in single-letter amino acid code. Colours highlight similarity of organization between species and indicates homology within, not between, species. Source data available: GEO: GSE112454

---

homogenization[23–25]. As tRNA genes may have performed important functions—as boundaries preventing heterochromatin spread into central cores and perhaps in higher-order centromere organization and architecture—tRNA gene clusters were maintained[26]. In *S. pombe*, non-centromeric and centromeric tRNA genes and 5S rDNAs cluster adjacent to centromeres in a TFIIIC-dependent manner[27,28]. The multiple tandem centromeric 5S rDNAs and tRNA genes could contribute to a robust, highly folded heterochromatin structure promoting optimum kinetochore configuration for co-ordinated microtubule attachments and accurate chromosome segregation[44].

The lack of overt sequence conservation between centromeres of different species appears not to prevent functional conservation, which may be driven by underlying sequence features or properties such as the transcriptional landscape. Maintenance of centromere function has been observed at a pre-established human centromere (pre-assembled with CENP-A and an intact, functional kinetochore) in chicken cells[45] (310 My divergence). Establishment of CENP-A chromatin on human alpha-satellite in mouse cells[46] (90 My divergence) is dependent on the 17 bp CENP-B box present at both human (alpha-satellite) and mouse (minor satellite) centromere sequences, and on the CENP-B protein. This conserved functionality is surpassed by the competence of *S. octosporus* central-core DNA to establish CENP-A chromatin in *S. pombe* from which it is separated by 119 My of evolution[20] (equivalent to 383 My using a chordate molecular clock) and lacks any clear conserved sequence elements akin to a CENP-B box. The analyses presented thus extend the evolutionary timescale over which cross-species establishment of CENP-A chromatin has been demonstrated.

## Methods

**Cell growth and manipulation**. Standard genetic and molecular techniques were followed. Fission yeast methods were as described[47]. Strains used in this study are listed in Supplementary Table 10. All *Schizosaccharomyces* strains were grown at 32 °C in YES (Yeast Extract with Supplements), except *S. cryophilus*, which was grown at 25 °C, unless otherwise stated. *S. pombe* cells carrying minichromosomes were grown in PMG-ade-ura. For low GFP-tagged CENP-A^Cnp1 protein expression from episomal plasmids, cells were grown in PMG-leu with thiamine.

**PacBio sequencing of genomic DNA**. High-molecular-weight genomic DNA was prepared from *S. cryophilus*, *S. octosporus* and *S. japonicus* using a Qiagen Blood and Cell Culture DNA Kit (Qiagen), according to the manufacturer's instructions. Pacific Biosciences (PacBio) sequencing was carried out at the CSHL Cancer Center Next Generation Genomics Shared Resource. Samples were prepared following the standard 20 kb PacBio protocol. Briefly, 10–20 μg of genomic material was sheared via g-tube (Covaris) to 20 kb. Samples were damage repaired via ExoVII (PacBio), damage-repair mix and end-repair mix using standard PacBio 20 kb protocol. Repaired DNA underwent blunt-end ligation to add SMRTbell adaptors. For some libraries, 10–50 kb molecules from 1 to 2 μg SMRTbell libraries were size selected using BluePippin (Sage Science), after which samples were annealed to Pacbio SMRTbell primers per the standard PacBio 20 kb protocol. Annealed samples were

sequenced on the PacBio RSII instrument with P4/C3 chemistry. Magbead loading was used to load each sample at a concentration between 50 and 200 pM. Additional PacBio sequencing (without BluePippin) was performed by Biomedical Research Core Facilities, University of Michigan. There, the following kits were used: DNA Sequencing Kit XL 1.0, DNA Template Prep Kit 2.0 (3 kb–10 kb) and DNA/Polymerase Binding Kit P4. MagBead Standard Seq v2 sequencing was performed using 10,000 bp size bin with no Stage Start with a 2 h observation time on a PacBio RSII sequencer. A summary of PacBio sequencing performed is listed in Supplementary Table 11.

**De novo whole genome assembly of PacBio sequence reads**. PacBio reads were assembled using HGAP3 (The Hierarchical Genome Assembly Process version 3)[48]. Reads were first sorted by length and the top 30% used as seed reads by HGAP3. All remaining reads of at least 1 kb in length were used to polish the seed reads. These polished reads were used to de novo assemble the genomes and Quiver software used to generate consensus genome contigs. Comparisons to the ChIP-seq input data and Broad Institute *Schizosaccharomyces* reference genomes[20] showed very high agreement with these datasets. The *S. octosporus* and *S. cryophilus* chromosomes were named according to their sequence lengths, the longest chromosome being labelled as chromosome I in each case.

**De novo assembly of *S. pombe* genome using nanopore technology**. Genomic DNA was extracted as described previously[49]. Briefly, cells were incubated with Zymolyase 20T to digest the cell wall, pelleted, resuspended in TE (10 mM Tris-HCl pH8, 1 mM EDTA) and lysed with SDS, followed by addition of potassium acetate and precipitation with isopropanol. After treatment with RNase A and proteinase K, two phenol chloroform extractions were performed. DNA was precipitated in the presence of sodium acetate and isopropanol, followed by centrifugation and washing of the pellet with 75% ethanol. After air drying, the pellet was resuspended in TE. DNA purity and concentration were assessed using a Nanodrop 2000 and the double-stranded high-sensitivity assay on a Qubit fluorometer, respectively. Genomic DNA was sequenced using the MinION nanopore sequencer (Oxford Nanopore Technologies). Three sequencing libraries were generated using the one-dimensional (1D) ligation kit SQK-LSK108, the two-dimensional (2D) ligation kit SQK-NSK007 and the 1D Rapid sequencing kit SQK-RAD002, following the manufacturer's guidelines. Each library was sequenced on one MinION flow cell. Sequencing reads were base-called using Metrichor (1D and 2D ligation libraries) or Albacore (rapid sequencing library). The combined dataset incorporating reads from three flow cells was assembled using Canu v1.5. The assembly was computed using default Canu parameters and a genome size of 13.8 Mbp. QUAST v3.2 was used to evaluate the genome assembly.

**Genome annotation and chromosome structure**. Genes were annotated onto the genome both de novo, using BLAST and the sequences of known genes, and by using liftover (https://genome-store.ucsc.edu) to carry over the previous gene annotation information from the Broad institute reference genomes (ref). Cross-Map[50] was then used to lift the chain files over to the new, updated genome. The locations of tRNA genes were predicted using tRNAscan[51,52]. Dfam 2.0[53] was used to annotate repetitive DNA elements. MUMmer3.23[54] was used to compare the genomes and annotate repeat elements and tandem repeat sequences, including those located in centromeric domain and telomere sequences. Centromeric repeat elements were assessed using ChIP-seq input data. Centromeric repeat elements were manually identified using BLASTN and MEGABLAST (https://blast.ncbi.nlm.nih.gov). Each repeat element was named according to their sequence features (association with tRNA gene and rDNAs) and locations. The sequence of the wild-type (h^90) *S. pombe* mating-type locus was obtained by manually merging nanopore and PacBio contigs using available data[20] (Supplementary Fig. 10) and information at www.pombase.org/status/mating-type-region. Genome synteny alignment analysis was carried out using syMAP42[55,56], based on orthologous genes among the three genomes.

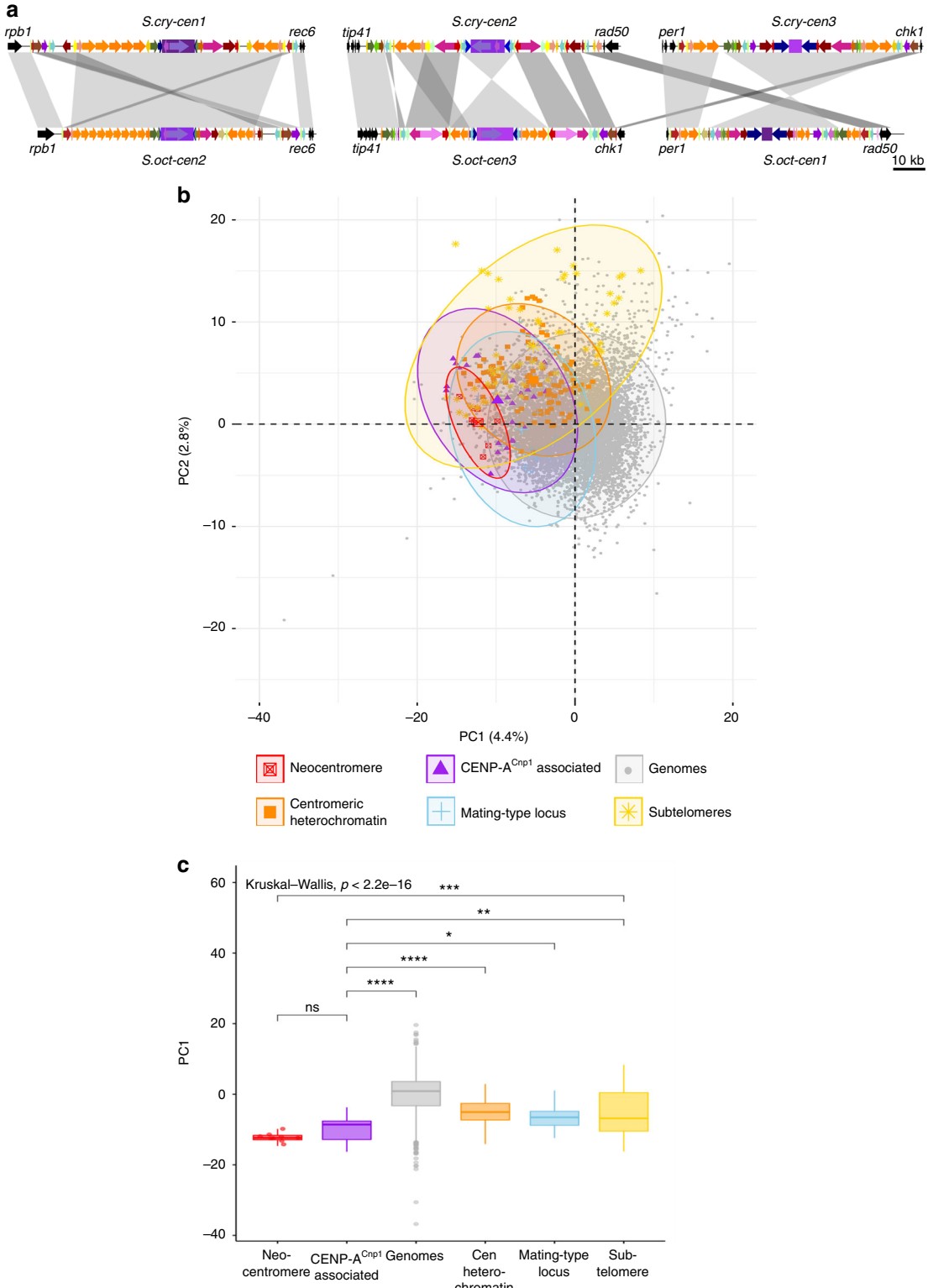

**Fig. 4** *Schizosaccharomyces* centromeres share ancestry and sequence features. **a** Structural alignment of putatively equivalent centromere repeat elements of *S. cryophilus* and *S. octosporus* to highlight potential centromere rearrangements during evolution. **b** Principal component analysis PC1 and PC2 of 5-mer frequencies of three fission yeast genomes. Genome regions (12 kb window) were assigned to one of five specific annotated groups (CENP-A$^{Cnp1}$-associated (purple, $n = 24$), centromeric heterochromatin (orange, $n = 112$), mating-type locus (blue, $n = 18$), subtelomeres (yellow, $n = 67$), neocentromere-forming regions[34] (red, $n = 9$), or other genome regions (grey, $n = 7652$). For each group the oval line encloses 95% of the data points. **c** Boxplot principal component PC1 of each group. Colours and values for $n$ as in **b**. Mean comparison between groups was used (*p*-value: > 0.05, ns; *$p > 0.01$; **$p > 0.001$; ***$p > 0.0001$; ****$p < 0.0001$)[66]. Centre line, medium; box limits, upper and lower quartiles; whiskers, 1.5 × interquartile range; points, outliers. Source data available: GEO: GSE112454.

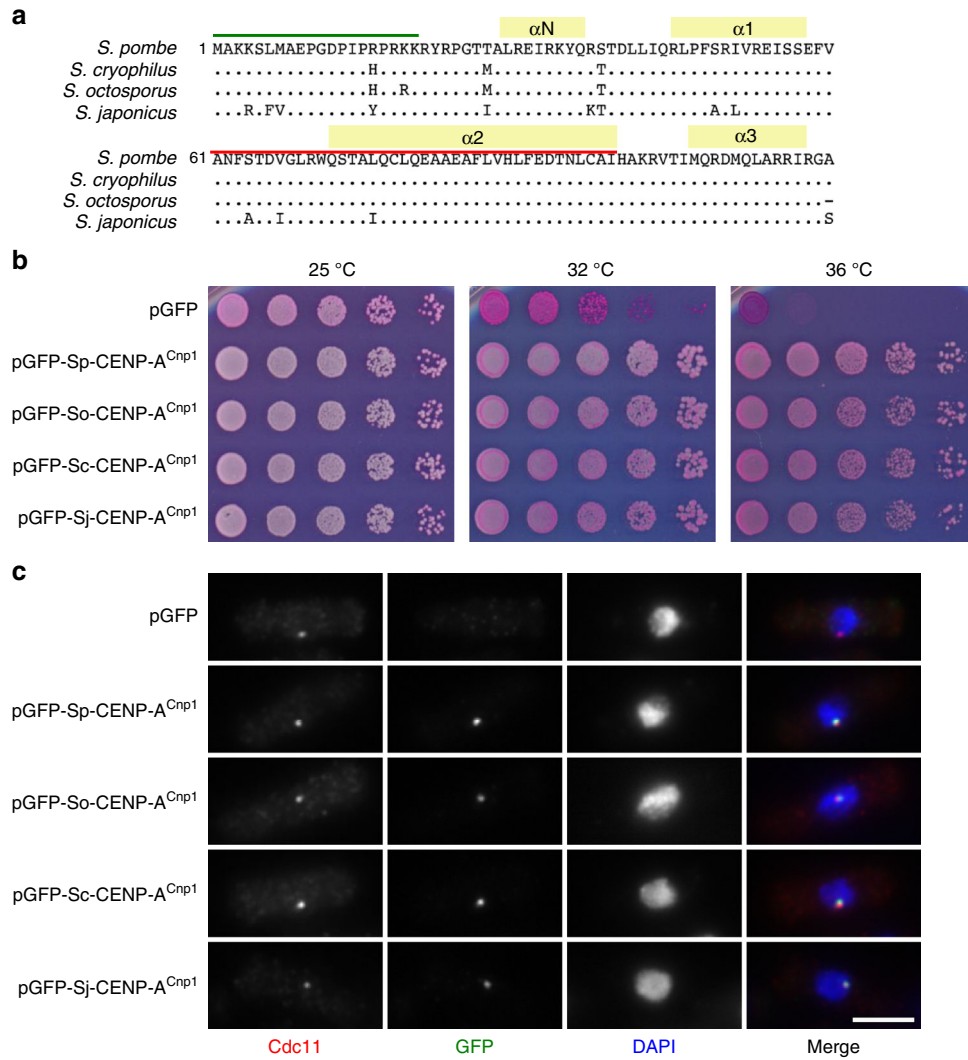

**Fig. 5** Cross-species functionality of CENP-A$^{Cnp1}$ proteins. **a** Alignment of *Schizosaccharomyces* CENP-A$^{Cnp1}$ proteins. Positions of alpha helices (yellow), N-terminal tail (green) and CENP-A-targeting domain (CATD; red) are indicated. **b** *S. pombe* temperature-sensitive *cnp1-1* cells expressing plasmid-borne GFP-CENP-A$^{Cnp1}$ from the indicated species (*Sp*, *S. pombe*; *So*, *S. octosporus*; *Sc*, *S. cryophilus*; *Sj*, *S. japonicus*), or GFP alone, were spotted on phloxine B-containing plates and incubated for 2–5 days at the indicated temperatures. **c** Localization of GFP-tagged CENP-A$^{Cnp1}$ from indicated *Schizosaccharomyces* species in *S. pombe*. Wild-type *S. pombe* cells bearing plasmids described in **a** were grown at 32 °C before fixation and staining with anti-GFP (green), anti-Cdc11 (red, spindle-pole body) and DAPI (blue, DNA). Centromeres cluster at the spindle-pole body in *S. pombe*. Scale bar, 5 μm. Source data available as a Source Data file

**ChIP-quantitative PCR analysis.** For analysis of CENP-A$^{Cnp1}$ association with minichromosomes bearing *S. octosporus* central-core DNA, three independent transformants with established centromere function (indicated by ability to form sectored colonies) for each minichromosome were grown in PMG-ade-ura cultures and fixed with 1% formaldehyde for 15 min at room temperature. ChIP was performed as previously described[57]. Briefly, $2.5 \times 10^8$ cells were lysed by bead beating (Biospec) in 300 μl Lysis Buffer (50 mM Hepes-KOH pH 7.5, 140 mM NaCl, 1 mM EDTA, 1% (v/v) Triton X-100, 0.1% (w/v) sodium deoxycholate). Lysates were sonicated (Bioruptor, Diagenode) for 20 min (30 s on/off, high setting), followed by centrifugation at $17,000 \times g$ ($2 \times 10$ min) to pellet cell debris. Lysates were precleared for 1 h with 25 μl of Protein-G agarose beads (Roche) and 10 μl precleared lysate retained as 'input' sample. Three hundred microlitres of lysate was incubated overnight with 10 μl sheep anti-CENP-A$^{Cnp1}$ serum and 25 μl Protein-G agarose beads. Beads were washed with Lysis Buffer, Lysis Buffer with 500 mM NaCl, WASH buffer (10 mM Tris-HCl pH 8, 0.25 M LiCl, 0.5% NP-40, 0.5% (w/v) sodium deoxycholate, 1 mM EDTA) and TE. DNA was recovered from input and IP samples using Chelex resin (BioRad). Ten microlitres of anti-CENP-A$^{Cnp1}$ sheep antiserum[57] (raised to the N-terminal 19 amino acids of *S. pombe* CENP-A$^{Cnp1}$) and 25 μl Protein-G-Agarose beads were used per ChIP. qPCR was performed using a LightCycler 480 and reagents (Roche), and analysed using Light-Cycler 480 Software 1.5 (Roche). Primers used in qPCR are listed in Supplementary Table 12. Mean %IP ChIP values for *Sp-cnt* or *So-cnt* on minichromsomes were normalized to %IP for endogenous *S. pombe* cnt1. Error bars represent SD.

**Chromatin immunoprecipitation sequencing.** A modified ChIP protocol was used. Briefly, pellets containing $7.5 \times 10^8$ cells were lysed by four 1 min pulses of bead beating in 500 μl of lysis buffer (50 mM HEPES-KOH, pH 7.5, 140 mM NaCl, 1 mM EDTA, 1% Triton X-100, 0.1% sodium deoxycholate), with resting on ice in between. The insoluble chromatin fraction was pelleted by centrifugation at $6000 \times g$ and washed with 1 ml lysis buffer before resuspension in 300 μl lysis buffer containing 0.2% SDS. Chromatin was sheared by sonication using a Bioruptor (Diagenode) for 30 min (30 s on/off, high setting). Nine hundred microlitres of lysis buffer (no SDS) was added and samples clarified by centrifugation at $17,000 \times g$ for 20 min and the supernatant used for ChIP. Six microlitres of anti-H3K9me2 mouse monoclonal mAb5.1.1[58] (kind gift from Takeshi Urano) or 30 μl sheep anti-CENP-A$^{Cnp1}$ antiserum[57] were used, along with protein-G-dynabeads (ThermoFisher Scientific) or Protein-G agarose beads (Roche), respectively. (For neocentromere strains, cells were first treated with Zymolyase 100T (AMS Biotechnology), washed in sorbitol and permeablized. Chromatin was fragmented with incubation with micrococcal nuclease. Cell suspensions were adjusted to standard ChIP buffer conditions and extracted chromatin was processed as per standard ChIP.) Immunoprecipitated DNA was recovered using Qiagen PCR purification columns. ChIP-Seq libraries were prepared with 1–5 ng of ChIP or 10 ng of input DNA. DNA was end-repaired using NEB Quick blunting kit (E1201L). The blunt, phosphorylated ends were treated with Klenow-exo$^-$ (NEB, M0212S) and dATP. After ligation of NEXTflex adaptors (Bioo Scientific) DNA was PCR amplified with Illumina primers for 12–15 cycles and library fragments of ~300 bp (insert plus

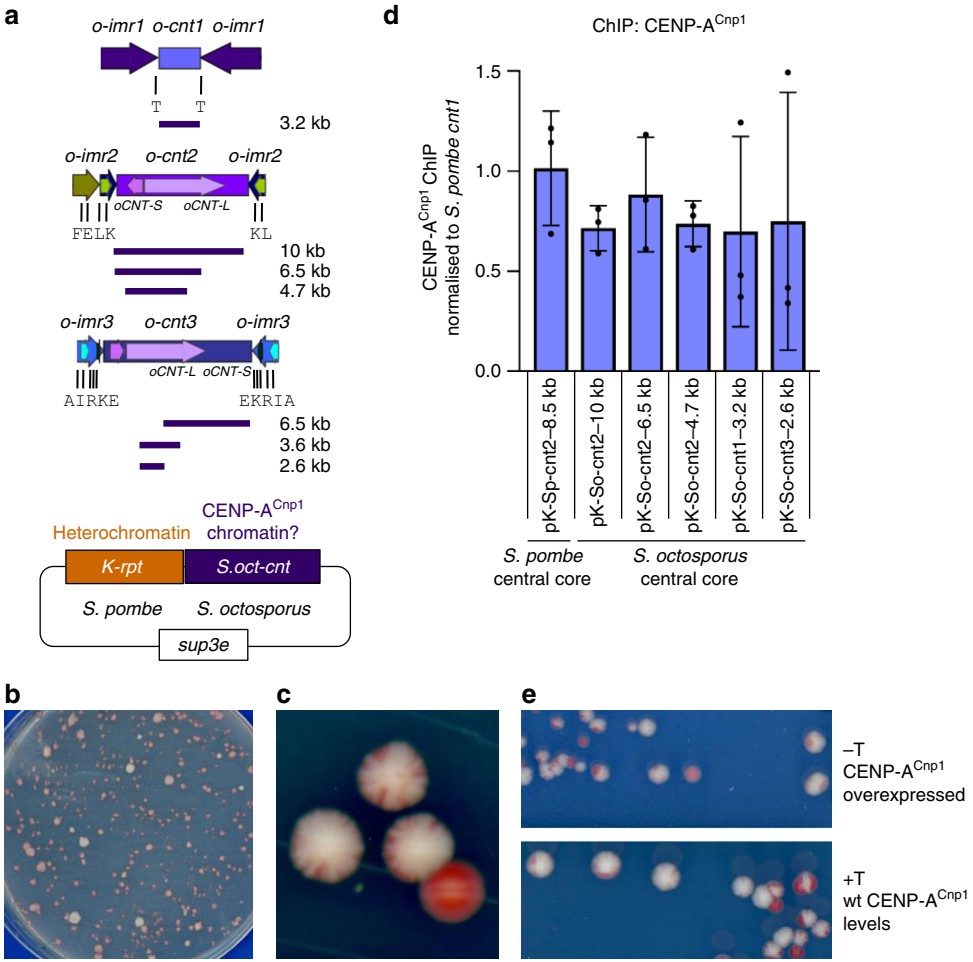

**Fig. 6** *S. octosporus* central-core DNA establishes CENP-A^Cnp1 chromatin upon introduction into *S. pombe*. **a** Indicated regions of *S. octosporus* central-core DNA placed adjacent to a portion of *S. pombe* heterochromatin-forming outer-repeat sequence on a plasmid. **b** *S. pombe* transformants containing minichromsome plasmids were replica-plated to low-adenine non-selective plates: colonies retaining the chimeric minichromosome plasmid are white/pale pink, those that lose it are red. Representative plate showing pKp-So-cnt3-6.5kb-containing colonies. **c** *S. pombe* cells containing pKp-So-cnt3-6.5 kb chimeric minichromosome were streaked to single colonies. Red colour indicates loss of minichromosome; small red sectors indicate low-frequency minichromosome loss and mitotic segregation function. **d** ChIP-qPCR for CENP-A^Cnp1 on *S. pombe* hi-CENP-A^Cnp1 cells containing chimeric minichromosomes with established centromere function. Three biologically independent transformants were analysed for each minichromosome (n = 3). ChIP enrichment on *S.pom-cnt2* and *S.oct-cnt*-bearing minichromosomes is normalized to the level at endogenous *S. pombe cnt1*. Individual data points are shown as black dots. Error bars, SD. **e** Propagation of chimeric minichromosome stability. Cells containing pK(5.6 kb)-So-cnt2-10 kb were streaked on low-adenine-containing plates with or without thiamine, which results in repression or expression of high levels of *S. pombe* CENP-A^Cnp1. Source data available as a Source Data file

adaptor sequences) were selected using Ampure XP beads (Beckman Coulter). The libraries were sequenced following Illumina HiSeq2000 work flow (or as indicated in Supplementary Table 11).

**Defining fission yeast centromeres**. CENP-A^Cnp1 and H3K9me2 ChIP-seq data were generated to identify centromere regions. ChIP-Seq reads with mapping qualities lower than 30, or read pairs that were over 500 nt or <100 nt apart, were discarded. ChIP-seq data were normalized with respect to input data. Paired-end ChIP-seq data (single-end for *S. japonicus*) was aligned to the updated genome sequences using Bowtie2[59]. Samtools[60], Deeptools[61] and IGV[62] were subsequently used to generate sequence data coverage files and to visualize the data. MACS2[63] was used to detect CENP-A^Cnp1 and heterochromatin-enriched regions of the genome.

**Centromere tRNA gene cluster analysis**. To test for the enrichment of tRNA gene clusters at centromere regions, a greedy search approach was used to identify potential clusters. All tRNA genes <1000 bp apart were grouped into clusters. To test for significant clustering of tRNA genes at the centromere, the locations of tRNA genes across the genome were shuffled 1000 times. For each cluster observed in the real genome, the proportion of permutations where the same cluster was observed at least as many times was calculated to provide estimates of significance. Following conversion of these *p*-values to *q*-values to

account for multiple testing, the centromere tRNA gene clusters each exhibited a *q*-value <0.005.

**Hsp16 gene tree analysis**. *hsp16* paralogs from *S. octosporus* and *S. cryophilus* genomes were predicted using BLASTP. The predicted protein sequences from *hsp16* genes across all four fission yeasts were aligned together with those from *Saccharomyces cerevisiae* using Clustal Omega. BEAST (Bayesian Evolutionary Analysis Sampling Trees)[64] and FigTree (http://tree.bio.ed.ac.uk/software/figtree/) was used to generate and view the *hsp16* gene phylogenetic tree.

**5-mer frequency PCA**. The CENP-A^Cnp1-associated sequences in the *S. pombe*, *S. cryophilus* and *S. octosporus* genomes are all ~12 kb in length. Each genome was therefore split into 12 kb sliding windows with a 4.5 kb overlap. The frequencies of each 5-mer was calculated in each window using Jellyfish[65]. CENP-A^Cnp1-associated regions showed a general enrichment of AT base pairs relative to the genome as a whole. To normalize for GC content among the windows, all base pairs were randomized in each sequence window to generate 1000 artificial sequences with the same GC content. 5-mer frequencies were then recalculated for each of these 1000 artificial sequences and the true original 5-mer frequencies compared with these background frequencies by calculating a *z*-score. Consequently, these enrichment scores represent the k-mer enrichments in a given sequence normalized for GC content. Genome windows were split into six groups: CENP-A^Cnp1-associated sequences (CENP-A^Cnp1 peaks covering >6 kb of sequence); outer-repeat

## Table 1 Establishment frequency and stability of minichromosomes in *S. pombe*

| Plasmid | Establishment frequency % (*n*) | Loss rate per division % (*n*) |
|---|---|---|
| pK-Sp-cnt2-8.5 kb | 94 (217) | 5.8 (3284) |
| pK-So-cnt2-10 kb | 36.4 (88) | 11.2 (1636) |
| pK-So-cnt3-6.5 kb | 40.1 (262) | 5.8 (3705) |
| pK-So-cnt2-6.5 kb | 28.1 (208) | 6.6 (3621) |
| pK-So-cnt2-4.7 kb | 5.2 (973) | 6.8 (7176) |
| pK-So-cnt1-3.2 kb | 1.9 (1529) | 11.5 (2017) |
| pK-So-cnt3-2.6 kb | 0.3 (1443) | 21.1 (1237) |
| pKp-So-cnt3-6.5 kb | 6.6 (916) | 15.9 (2099) |
| pKp-So-cnt3-3.6 kb | 0 (1538) | NA |
| pKp | 0 (295) | NA |

Establishment frequency of chimeric minichromosomes in *S. pombe* hi-CENP-A[Cnp1] cells determined by replica plating of transformants (Methods) as shown in Fig. 6b (*n* = number of transformants analysed). Chromosome loss rate of established minichromosome was determined by half-sector assay (Methods). Two transformants containing established centromeres were analysed for each minichromosome and the mean loss rate determined, *n* = number of colonies screened. NA, not applicable as the minichromosomes did not establish centromere function

heterochromatin regions (more than half the window covered by H3K9me2 peaks adjacent to CENP-A domains); subtelomeric regions (more than half the window covered by H3K9me peaks and close to the end of a chromosome); mating-type locus; neocentromere regions (identified using CENP-A[Cnp1] ChIP-seq data of *S. pombe* neocentromere-containing strains[34]); and remaining genome sequences. As the highly repetitive transposon-rich *S. japonicus* centromere regions are not fully assembled, the precise location of the centromere-kinetochore is unknown. We therefore adopted a limited PCA approach and selected the top 11 most highly enriched 12 kb regions from CENP-A[Cnp1] ChIP-seq (Supplementary Data 11). These were compared with ten randomly selected non-centromeric sequences from each of the fission yeast genomes as above.

Logistic regression and mean comparison were used to determine whether principal components were linked to the probability of a sequence belonging to a particular sequence group[66]. Logistic regression and mean comparison were used to determine whether principal components (FactoMineR) were linked to the probability of a sequence belonging to a particular sequence group.

**Construction of minichromosomes**. *S. pombe* functional minichromosomes contain central domain DNA and flanking repeat DNA on one side; the long, inverted repeats found in the natural context are not tolerated by *Escherichia coli*[67]. Regions of *S. octosporus* and *S. cryophilus* central-core regions were amplified with primers indicated in Supplementary Table 12 and inserted as *Bgl*II-*Nco*I, *Bam*HI-*Nco*I or *Bgl*II-*Sal*I fragments into *Bgl*II-*Nco*I- or *Bgl*II-*Sal*I-digested plasmid pK (5.6 kb)-MCS-ΔBam, which contains a 5.6 kb fragment of the *S. pombe* K (*dg*) outer repeat. To create plasmid pK-So-cnt2-10 kb, an additional 3.6 kb region from *S.oct-cnt2* was inserted as a *Bam*HI-*Sal*I fragment into *Bgl*II-*Sal*I-digested pK-So-cnt2-6.5 kb to make a 10 kb region of *S. octosporus* central core. For pKp plasmids, *S. octosporus* or *S. cryophilus* central-core regions were by inserted as *Bgl*II-*Nco*I, *Sal*I-*Bam*HI or *Xho*I-*Bam*HI fragments into *Bam*HI-*Nco*I- or *Sal*I-*Bam*HI-digested plasmid pKp (pMC91), which contains 2 kb region from *S. pombe* K(*dg*) outer repeat. For the *S. japonicus* CENP-A[Cnp1]-associated retrotransposon Tj7[20] (Supplementary Fig. 6), a region spanning the almost the entire retrotransposon (but lacking the second LTR to avoid rearrangement or transposition problems in *E. coli* or *S. pombe*) was amplified by PCR with primers indicated in Supplementary Table 12 and cloned in two steps into the *Not*I-*Xba*I site of pK(5.6 kb)-MCS-ΔBam to make pK-Sj-Tj7-4.8 kb. Plasmids are listed in Supplementary Table 13.

**Centromere establishment assay**. Strains A7373 or A7408, which contains integrated *nmt41*-GFP-CENP-A[Cnp1] to allow high level expression of CENP-A[18], were grown in PMG-complete medium and transformed using sorbitol-electroporation method[68]. Cells were plated on PMG-uracil-adenine plates and incubated at 32 °C for 5–10 days until medium-sized colonies had grown. Colonies were replica-plated to PMG-low-adenine (10 μg/ml) plates to determine the frequency of establishment of centromere function. These indicator plates allow minichromosome loss (red) or retention (white/pale pink) to be determined. Minichromosome retention indicates that centromere function has been established, and that minichromosomes segregate efficiently in mitosis. In the absence of centromere establishment, minichromosomes behave as episomes that are rapidly lost. Minichromosomes occasionally integrate giving a false positive white phenotype. To assess the frequency of such integration events and to confirm establishment of centromere segregation function, a proportion of colonies giving the white/pale pink phenotype upon replica plating were re-streaked to single colonies on low-adenine plates—sectored colonies are indicative of segregation function with low levels of minichromosome loss, whereas pure white colonies are indicative

of integration into endogenous chromosomes—and the establishment frequency adjusted accordingly.

**Minichromosome stability assay**. Minichromosome loss frequency was determined by half-sector assay. Briefly, transformants containing minichromsomes with established centromere function were grown in PMG-ade-ura to select for cells containing the minichromosome. At least two transformants were analysed per minichromosome. Cells were plated on low-adenine-containing plates and allowed to grow non-selectively for 4–7 days. Minichromosome loss is indicated by red sectors and retention by white sectors. To determine loss rate per division, all colonies were examined with a dissecting microscope. All colonies—except pure reds—were counted to give total number of colonies. Pure reds were checked for the absence of white sectors and were excluded, because they had lost the minichromosome before plating. To determine colonies that lost the minichromosome in the first division after plating, 'half-sectored' colonies were counted. This included any colony that was 50% or greater red (including those with only a tiny white sector). Loss rate per division is calculated as the number of half-sectored colonies as a percentage of all (non-pure-red) colonies.

**Recovery of minichromosomes from *S. pombe***. To confirm that establishment of centromere function by minichromosome plasmids was not due to rearrangement or gain of *S. pombe* central-core sequences, minichromosomes were recovered from *S. pombe* into *E. coli*. Approximately $1 \times 10^8$ *S. pombe* cells containing minichromosome plasmids with established centromere function were incubated 1 ml PEMS buffer (100 mM PIPES pH 7, 1 mM EDTA, 1 mM MgCl$_2$, 1.2 M Sorbitol) containing 1 mg/ml Zymolyase-100T (AMS Biotechnology) for 30–60 min at 36 °C to digest cell walls. After pelleting and washing with PEMS, spheroplasts were lysed and plasmid DNA isolated using Qiagen miniprep kit, following manufacturer's instructions. Due to low-yield plasmids were recovered by transformation into *E. coli* gt116 cells, followed by restriction enzyme analysis of resultant miniprep DNA. Digestion patterns of recovered and original minichromsome plasmids was compared by agarose gel electrophoresis.

**Immunolocalization**. For localization of CENP-A[Cnp1], *Schizosaccharomyces* cultures were fixed with 3.7% formaldehyde for 7 min, before processing for immunofluorescence as described[57]. Briefly, cells were fixed with 3.7% formaldehyde for 7 min, followed by cell-wall digestion with Zymolyase-100T (AMS Biotechnology) in PEMS buffer (100 mM PIPES pH 7, 1 mM EDTA, 1 mM MgCl$_2$, 1.2 M Sorbitol). After permeablization with Triton X-100, cells were washed, blocked in PEMBAL (PEM containing 1% bovine serum albumin, 0.1% sodium azide, 100 mM lysine hydrochloride). Anti-CENP-A[Cnp1] sheep antiserum[57] (raised to the N-terminal 19 amino acids of *S. pombe* CENP-A[Cnp1]) was used in PEMBAL at 1:1000 dilution and Alexa-488-coupled donkey anti-sheep secondary antibody (A11015; Invitrogen) at 1:1000 dilution. Cells were stained with 4′,6-diamidino-2-phenylindole (DAPI) and mounted in Vectashield. Microscopy was performed with a Zeiss Imaging 2 microscope (Zeiss) using a × 100 1.4 NA Plan-Apochromat objective, Prior filter wheel, illumination by HBO100 mercury bulb. Image acquisition with a Photometrics Prime sCMOS camera (Photometrics, https://www.photometrics.com) was controlled using Metamorph software (Universal Imaging Corporation). Exposures were 1500 ms for FITC/Alexa-488 channel and 300–1000 ms for DAPI. Images shown in Fig. 2a are autoscaled.

To express GFP-tagged versions of *Schizosaccharomyces* CENP-A[Cnp1] proteins in *S. pombe*, ORFs were amplified from relevant genomic DNA using primers listed in Supplementary Table 12. Fragments were digested with *Nde*I-*Bam*HI or *Nde*I-*Bgl*II and ligated into *Nde*I-*Bam*HI digested pREP41X-GFP vector[69] (Supplementary Table 13). For detection of GFP-tagged versions of *Schizosaccharomyces* CENP-A[Cnp1] proteins in *S. pombe*, cells containing pREP41X-GFP-CENP-A[Cnp1] episomal plasmids (variable copy number) were grown in PMG-leu + thiamine to allow low GFP-CENP-A[Cnp1] expression. Cells were fixed, processed for immunolocalization and microscopy as above. Anti-GFP antibody (A11122; Invitrogen) was used at 1:300 and anti-Cdc11[57] (a spindle-pole body marker; gift from Ken Sawin) was used at 1:600. Secondary antibodies were, respectively, Alexa-488-coupled chicken anti-rabbit (A21441; Invitrogen) and Alexa-594-coupled donkey anti-sheep (A11016; Invitrogen), both at 1:1000. Exposures were FITC/488 channel 1500 ms, TRITC/594 1000 ms and DAPI 500–1000 ms. For display of images in Fig. 5C, TRITC/594 and FITC/488 images are scaled relative to the maximum intensity in the set of images, whereas DAPI images are autoscaled.

**Reporting summary**. Further information on research design is available in the Nature Research Reporting Summary linked to this article.

## Data availability

Sequence data generated in this study have been submitted to GEO under accession number: GSE112454. This study used PacBio and nanopore sequencing data under project PRJNA472404. Assembled genomes are available at http://bifx-core.bio.ed.ac.uk/~ptong/genome_assembly/. All other relevant data supporting the key findings of this study are available within the article and its Supplementary Information files or from the corresponding authors upon reasonable request. The source data underlying Figs. 2, 5, 6 and Supplementary Figs 6, 8, 9, are provided in a Source Data file. A reporting summary for this Article is available as a Supplementary Information file.

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

## Acknowledgements

We thank Alastair Kerr, Shaun Webb and Daniel Robertson for bioinformatics support, David Kelly for microscopy support, Ken Sawin and Takeshi Urano for antibodies, and Kojiro Ishii, Ken Sawin and Nick Rhind for yeast strains. We thank Robert Lyons, Joe Washburn, Christina McHenry (University of Michigan) and Greg J. Hannon, Richard McCombie, Eric Antoniou and Sara Goodwin (CSHL) for PacBio sequencing. We are grateful to Chris Ponting for advice and comments on the manuscript and to Sandra Catania and other members of the Allshire and Heun labs for helpful discussions. N.R.T.T., R.A. and J.T.-G. were supported by the Darwin Trust of Edinburgh. The Darwin Trust and a Principal's Career Development scholarship supported N.R.T.T. P.T. was partly supported by funding from the European Commission Network of Excellence EpiGeneSys- (HEALTH-F4-2010-257082) and a Wellcome Enhancement Award (095021) to R.C.A. R.C.A. is a Wellcome Principal Research Fellow (095021, 200885); the Wellcome Centre for Cell Biology is supported by core funding from Wellcome (203149). C.A.M. and C.A.N. are supported by Biotechnology and Biological Sciences Research Council (BBSRC) grant BB/N016858/1 and Wellcome Investigator Award 110064/Z/15/ Z. Pacific Biosciences (PacBio) sequencing carried out at the CSHL Cancer Center Next Generation Genomics Shared Resource, which is supported by the Cancer Center Support Grant 5P30CA045508 was paid for by a kind gift from Kathryn W. Davis to G.J.H.

## Author contributions

R.C.A. and A.L.P. designed the study. P.T. performed the PacBio genome assemblies and bioinformatics, ChIP-seq analysis and PCA analysis. C.M. performed the nanopore sequencing of *S. pombe* supervised by C.N. H.B., N.R.T.T., J.T.-G. and R.A. generated ChIP-seq data with contribution from M.S. A.L.P. performed cytology, analysis of repetitive regions and experiments on cross-species functionality. R.C.A. supervised the study. A.L.P. wrote the manuscript with contributions from P.T., R.C.A. and other authors. All authors read and approved the final version of the manuscript.

## Additional information

**Competing interests:** The authors declare no competing interests.

