## [Peer Review File · Nature Communications]

Reviewers' comments:

Reviewer #1 (Remarks to the Author):

The manuscript by Tong, Pidoux, et al. investigates the conservation of the centromere sequence between three *Schizosaccharomyces* species. They first used long-read sequencing to extend the genome sequences of *S. octosporus* and *S. cryophilus*. They found that that *Schizosaccharomyces* centromeres lack sequence homology but do have conserved synteny and CENP-A central domains separated by clusters of tRNA genes assembled in heterochromatin. *S. japonicas* did not have the conserved synteny. *S. octosporus* and *S. cryophilus* have 5SrRNA genes in heterochromatic outer-repeats but *S. pombe* does not. tDNAs or an LTR occur at the transition between CENP-A and heterochromatin and may act as boundaries for the heterochromatin. CENP-A protein from *S. octosporus*, *S. cryophilus*, and *S. japonicus* can complement a *S. pombe* CENP-A ts mutant and can integrate into DNA. Fusion of a *S. octosporus* central core with a *S. pombe* outer repeat on a minichromosome allowed establishment of the CENP-A chromatin and plasmid segregation.

Overall, this manuscript contains a wealth of sequencing data, especially the more difficult regions to sequence due to their repetitive nature. This data should be extremely useful to the community. In addition, this study will be a significant addition to the chromosome segregation field with the experiments highlighted below.

The comparison of centromeres among *Schizosaccharomyces* species is interesting. However, it is unclear why many of the studies lacked *S. japonicus*, which was used in some experiments but not others. Given that this species has some very divergent aspects, it would be an important comparison.

The main conclusion from this study is that although centromeres from different *Schizosaccharomyces* lack sequence conservation, there are conserved properties that allow the establishment and of function of assembly of CENP-A chromatin to be recognized by other species. This conclusion is based on the studies using the *S. octosporus* central core placed on a minichromosome next to *S. pombe* outer repeats and transformed into *S. pombe* to test CENP-A assembly and retention of the plasmid without selection. These experiments are convincing that the *S. octosporus* central core contains conserved properties, but it is unclear if the other two species will give the same result. Therefore, it is unclear if the statement of the main conclusion is generalizable for all *Schizosaccharomyces* species. The functional experiments of all three other *Schizosaccharomyces* species is crucial. Even if they do not give the tantalizing result of establishing CENP-A chromatin in *S. pombe*, it could help illustrate which conserved elements are important.

Reviewer #2 (Remarks to the Author):

This paper addresses the question of centromere identity from the DNA sequence point of view. Centromere identity is dictated by the specific deposition of CENP-A, while centromeric DNA sequences are not conserved between species. However, understanding the contribution of the DNA sequence to the establishment of CENP-A chromatin and the mechanisms involved in its centromere specific localization is of high interest to the centromere field. In this paper, authors have assembled the genomes of *Schizosaccharomyces octosporus* and *S. cryophilus* and have performed a solid bioinformatics analysis to study genome organization and synteny analysis in *Schizosaccharomyces*, comparing *S. octosporus*, *S. cryophilus* and *S. pombe*. They observed that, although DNA sequence is not conserved, centromeres from the three species share a common domain organization with a central domain containing CENP-ACnp1 flanked by heterochromatic outer-repeat regions. The presence of tRNA and 5S rRNA genes is also a common feature in all three species, with *S. octosporus* and *S. cryophilus* showing conserved syntenic clusters. Though

robust, these results are mainly descriptive/confirmatory.

The authors also addressed the question of functional conservation, showing that CENP-ACnp1 from *S. octosporus*, *S. cryophilus* and *S. japonicus* localize to centromeres in *S. pombe* and complemented the *S. pombe* temperature sensitive *cnp1-1* mutation. Moreover, they perform PC-analysis of 5-mer frequencies to show that, despite the lack of sequence conservation, the central-core sequences of the three species share common features. Finally, authors convincingly show that the central-core region from *S. octosporus* is functional in *S. pombe*, being able to incorporate CENP-ACnp1 and establish a functional centromere. From these results the authors propose that centromeric DNAs have intrinsic features that are conserved through evolution and promote centromeric chromatin assembly. What are these features? What is the mechanism by which they dictate CENP-A assembly? These are questions that are not even discussed. From this point of view, this work fails short since functional conservation has already been shown between human and mice and chicken. I would expect that the authors would have addressed these very important questions.

Some minor points are:

- Figure legend Figure S1d: "Blue shading indicates homologous genes between species". Blue is grey.
- Figure legend Figure S4c: "Purple triangle indicates position of *S.oct-cen*" Is not *S.cry-cen*? "Homologies lies within cTART14 elements (also present at *S. cryophilus* *cen1* and *cen3*)". Should be "*cen1* and *cen2*"

Reviewer #3 (Remarks to the Author):

This manuscript reports on the structure and organization of centromeres in related fission yeasts *S. pombe*, *S. octosporus*, and *S. cryophilus* (and *S. japonicus*). The authors use long read DNA sequencing to fully assemble each genome and repetitive centromeric sequences and combine this with ChIP-seq CENP-A and H3K9me2 localization, and functional analysis. Their results reveal the conserved and diverged features of centromere organization including highly conserved syntenic 5S RNA and tRNA genes across centromeres with the latter appearing to play a conserved boundary role. The authors also conclude that non-homologous central core sequences from *S. Octosporus* and *S. pombe* play conserved roles in de novo centromere establishment. These results are very interesting and of broad general interest. Although the results are mostly descriptive, I really enjoyed reading this paper and have only one concern noted below regarding the second conclusion.

I think the authors are overstating the case for cross-species conservation of cen DNA features that promote CENP-A assembly. Their data only show that *S. octosporus* cnt DNA is permissive for CENP-A assembly when placed next to *S. pombe* outer cen repeat sequences that can promote CENP-A assembly and centromere function. The underlying sequence has not been demonstrated to have de novo centromere assembly function. The authors previous work has shown that other non-related sequences can also be permissive for CENP-A assembly in this assay. The authors should tone down this point in both their abstract and at the end of the text.

The conservation of tDNA and 5S gene arrangements described by the authors are particularly interesting. This paper will become a valuable resource to the community and a starting point for further analysis of these and other centromere feature revealed by the results.

Reviewer #4 (Remarks to the Author):

This is a very well written manuscript that was a real pleasure to review. In brief, it describes the assembly of three *Schizosaccharomyces* genomes with a focus on the repetitive genome components specifically of the centromeres and telomeres. The quality of the figures is similarly exceptional. In short, it is a great paper and my comments are minor in scope and intended to clarify a few points for the non-expert reader.

Key claims of the manuscript include 1) conservation, over millions of years, of syntenic clusters of tRNA and rRNA genes and 2) a non-homologous centromere core sequence from one species providing centromere function in another. The claims are well substantiated with the caveats listed below, which might justify minor changes or additions to the discussion.

The syntenic arrangement of tRNA genes near *S.cry-cen1* and 2 and *S.oct-cen2* and 3 together with the CENP-A and H3K9me2 ChIP data does suggest that they constitute a barrier to heterochromatin spread as suggested by the authors. However, the authors do not comment on the two CENs that are not flanked by tRNAs are the only ones composed of inverted repeats (*S.cry-cen3* and *S.oct-cen1*). LTRs are indicated for *S.cry-cen3* as fulfilling this “demarcation” function, but no such LTRs are marked in the apparently paralogous *S.oct-cen1* (Fig. 2b/c). Thus the inverted nature of both of these CENs seems to me to be at least as important as the presence of flanking LTRs in one of them.

The experiments describing the functioning of the So CNT in *S. pombe* are fascinating and very nicely done. In light of the K-mer analysis, it would be nice to know how critical the CNT repeats are, i.e. can any DNA consisting of “CENP-A-favored” kmers and flanked by the outer repeats function as centromere? Do previous publications exist where such swaps have been attempted? Slightly more discussion of why the “NA” and “ND” clones did not work (see comment below regarding Figure 6d) could be enlightening. A little more background regarding these experiments might also be useful, e.g.: Why was only one *S. pombe* outer repeat used to test So-cnt in *S. pombe*, when the actual *S. pombe* centromeres are flanked by inverted repeats? Is this a technical limitation of cloning inverted repeats in *E. coli* or *S. cerevisiae*? What do the circles above the “K-rpt” and “S.oct-cnt” in Figure 6a indicate (nucleosome positions)? And is the “K-rpt” the same sequence that is orange in Fig. S5A? IF not, it might be worth highlighting that particular repeat in Fig. S5A.

Minor comments:

The authors may need to define the term “tDNA”, which I suspect is shorthand for tRNA genes?

Page 5: “benefitting chromosome segregation selected” – comma before “selected”

Page 5: “TFIIIC” – may need definition, though easily googled

Page 15: Figure 4a legend needs a period

Figure 6d: “ND” and “NA” are not defined in figure or legend. It is critical to know what these mean in order to evaluate the results.

Figure S1 legend: “Blue shading” should be “Grey shading”?

Figure S3A legend: Mention that black circles = centromeres, as not listed in Figure.

Reviewer #1 (Remarks to the Author):

The manuscript by Tong, Pidoux, et al. investigates the conservation of the centromere sequence between three Schizosaccharomyces species. They first used long-read sequencing to extend the genome sequences of *S. octosporus* and *S. cryophilus*. They found that that Schizosaccharomyces centromeres lack sequence homology but do have conserved synteny and CENP-A central domains separated by clusters of tRNA genes assembled in heterochromatin. *S. japonicus* did not have the conserved synteny. *S. octosporus* and *S. cryophilus* have 5SrRNA genes in heterochromatic outer-repeats but *S. pombe* does not. tDNAs or an LTR occur at the transition between CENP-A and heterochromatin and may act as boundaries for the heterochromatin. CENP-A protein from *S. octosporus*, *S. cryophilus*, and *S. japonicus* can complement a *S. pombe* CENP-A ts mutant and can integrate into DNA. Fusion of a *S. octosporus* central core with a *S. pombe* outer repeat on a minichromosome allowed establishment of the CENP-A chromatin and plasmid segregation.

Overall, this manuscript contains a wealth of sequencing data, especially the more difficult regions to sequence due to their repetitive nature. This data should be extremely useful to the community. In addition, this study will be a significant addition to the chromosome segregation field with the experiments highlighted below.

We thank the reviewer for her/his comments. Below we describe the additional information included in our revised manuscript.

The comparison of centromeres among Schizosaccharomyces species is interesting. However, it is unclear why many of the studies lacked *S. japonicus*, which was used in some experiments but not others. Given that this species has some very divergent aspects, it would be an important comparison.

The repetitive nature of *S. japonicus* centromere regions has so far hindered complete assemblies across these centromeres. Thus, analyses for *S. japonicus* are subject to the caveat that the organisation of repeats at the *bona fide* centromere (i.e. where the kinetochore assembles) remains to be determined. However, we have now carried out limited analyses of *S. japonicus* CENP-A-associated regions (described below).

The main conclusion from this study is that although centromeres from different Schizosaccharomyces lack sequence conservation, there are conserved properties that allow the establishment and of function of assembly of CENP-A chromatin to be recognized by other species. This conclusion is based on the studies using the *S. octosporus* central core placed on a minichromosome next to *S. pombe* outer repeats and transformed into *S. pombe* to test CENP-A assembly and retention of the plasmid without selection. These experiments are convincing that the *S. octosporus* central core contains conserved properties, but it is unclear if the other two species will give the same result. Therefore, it is unclear if the statement of the main conclusion is generalizable for all Schizosaccharomyces species. The functional experiments of all three other Schizosaccharomyces species is crucial. Even if they do not give the tantalizing result of establishing CENP-A chromatin in *S. pombe*, it could help illustrate which conserved elements are important.

We have now performed similar cross-species analyses for *S. cryophilus* centromere sequences as were performed for *S. octosporus* centromere sequences in the original manuscript. For *S. cryophilus* we inserted regions from *Sc-cnt1* (homologous to *Sc-cnt2*) and *Sc-cnt3* into a plasmid containing *S. pombe* outer repeat. The resulting minichromosome plasmids were able to establish centromere segregation function and assemble CENP-A chromatin in *S. pombe* cells (albeit with a lower establishment frequency than *S. pombe* and *S. octosporus* sequences). This new data is presented in Supplementary Figure S8 and mentioned in the text.

For *S. japonicus*, only limited analyses were possible. We took the region upon which the highest levels of CENP-A^{Cnp1} are detected by ChIP-seq - retrotransposon Tj7 – and constructed a similar minichromosome plasmid. Only one LTR was included to guard against potential plasmid

rearrangement and transposition activity in *S. pombe*. This minichromosome did not form functional centromeres when introduced into *S. pombe*. Only 2 potential candidates out of ~4000 transformants presented colony colour suggestive of centromere function. Moreover, ChIP-qPCR of these two candidates indicated CENP-A^{Cnp1} chromatin was not appreciably assembled on Tj7 in *S. pombe*. This data is now presented in Supplementary Fig. 6. Further, more comprehensive investigations, perhaps with using a combination of different retrotransposon sequences, may be required. However, it would be prudent to first determine what *S. japonicus* centromere sequences are sufficient for centromere establishment in *S. japonicus* itself. Such experiments are more long term and thus beyond the scope of this study.

Because *S. japonicus* centromere regions remain to be fully assembled we do not know the *bona fide* site of kinetochore assembly. ChIP-seq reads from repeats map to all copies of a repeat. The highest CENP-A^{Cnp1} levels might be assembled upon regions with a distinctive organisation of multiple Tj7 (and Tj6) repeats of which we remain unaware. We don't know, for instance, whether a single Tj7 repeat really does assemble CENP-A^{Cnp1} in *S. japonicus*.

We have also performed Principal Component Analysis on the *S. japonicus* genome. Whereas the CENP-A-associated regions of *S. pombe*, *S. cryophilus* and *S. octosporus* all cluster together in PCA for 5-mers, *S. japonicus* CENP-A-associated regions (Tj7 and Tj6) do not share these common features. This data is presented in Supplementary Fig. S6.

Thus, three species share similar centromere organisation and features (Figs 2-4) and show cross-species functionality (Fig 6, Supplementary Fig S8), whilst *S. japonicus* has distinct centromere organisation and properties and Tj7, the most highly CENP-A-associated region in the genome (so far identified) appears to not exhibit cross-species functionality in *S. pombe* (Supplementary Fig. S6)

Reviewer #2 (Remarks to the Author):

This paper addresses the question of centromere identity from the DNA sequence point of view. Centromere identity is dictated by the specific deposition of CENP-A, while centromeric DNA sequences are not conserved between species. However, understanding the contribution of the DNA sequence to the establishment of CENP-A chromatin and the mechanisms involved in its centromere specific localization is of high interest to the centromere field. In this paper, authors have assembled the genomes of *Schizosaccharomyces octosporus* and *S. cryophilus* and have performed a solid bioinformatics analysis to study genome organization and synteny analysis in *Schizosaccharomyces*, comparing *S. octosporus*, *S. cryophilus* and *S. pombe*. They observed that, although DNA sequence is not conserved, centromeres from the three species share a common domain organization with a central domain containing CENP-ACnp1 flanked by heterochromatic outer-repeat regions. The presence of tRNA and 5S rRNA genes is also a common feature in all three species, with *S. octosporus* and *S. cryophilus* showing conserved syntenic clusters. Though robust, these results are mainly descriptive/confirmatory.

The analyses we have presented provide the complete assembly of centromeres in two additional fission yeast species and partial assembly of centromeres in a third species along with detailed annotation. In addition, the functional analyses presented show that non-homologous sequences from the centromeres two species can function in a third species. These functional tests indicate that cryptic embedded sequence features allow recognition of these non-homologous centromeres DNA to enable CENP-A and kinetochore assembly.

The authors also addressed the question of functional conservation, showing that CENP-ACnp1 from *S. octosporus*, *S. cryophilus* and *S. japonicus* localize to centromeres in *S. pombe* and complemented the *S. pombe* temperature sensitive *cnp1-1* mutation. Moreover, they perform PC-analysis of 5-mer frequencies to show that, despite the lack of sequence conservation, the central-core sequences of the three species share common features. Finally, authors convincingly show that the central-core region from *S. octosporus* is functional in *S. pombe*, being able to incorporate CENP-ACnp1 and establish a functional centromere. From these results the authors propose that centromeric DNAs have intrinsic features that are conserved through evolution and promote centromeric chromatin assembly. What are these features? What is the mechanism by which they dictate CENP-A assembly? These are questions that are not even discussed. From this point of view, this work fails

short since functional conservation

has already been shown between human and mice and chicken. I would expect that the authors would have addressed these very important questions.

We agree with the reviewer that these are key questions and it will be of great interest to discover precisely what features and processes are critical in establishment of CENP-A chromatin. We have discussed these questions at length in our previous publications to which we refer. We have added additional text to the Discussion describing some of the possible features that might contribute competence of centromere DNA in establishing CENP-A chromatin. Future work, beyond the scope of this manuscript, will be required to tease out which functional, conserved aspects of central core sequences are key in determining the ability to establish CENP-A chromatin. Rather than there being one key feature we suspect that multi-fold redundancy is involved, hence it is difficult to define a precise mechanism that mediates CENP-A, rather than H3, assembly on these centromeric DNAs.

Although cross-species functionality has been demonstrated between mouse and human centromere DNA, human alpha-satellite and mouse minor satellite share a 17-bp sequence – the CENP-B box. Competence to establish CENP-A chromatin in mammalian cells is dependent on the CENP-B box and on the CENP-B protein (although neither are required for maintenance of CENP-A chromatin) (Okada et al, 2007). In the case of chicken and human, somatic cell fusion was used, i.e. the human centromeres already had CENP-A and a fully functioning kinetochore so this in fact tested only maintenance (albeit with chicken CENP-A), not establishment. We have stated this more explicitly in the final paragraph. As has been demonstrated through CENP-A/HJURP tethering experiments, once established, CENP-A chromatin can be maintained on non-centromeric DNA (Mendiburo et al, 2011; Barnhart et al 2011). Thus, our observations with *Schizosaccharomyces* centromere DNA are notable for the following reasons: the evolutionary distance over which centromere function is established on heterologous DNA is greater than in previous examples; these non-homologous sequences show cross-species establishment of CENP-A chromatin (not just maintenance); no conserved distinct DNA-binding protein component or element such as the 17-bp CENP-B box is evident at *Schizosaccharomyces* centromeres. In contrast to mammalian centromeres, there is no overt sequence homology between the *Schizosaccharomyces* centromeres, only an underlying similarity revealed by PCA of 5-mers (Fig 4) that may be indicative of underlying features and processes. It is therefore our opinion that these data are striking even in comparison to the previous observations

Some minor points are:

- Figure legend Figure S1d: “Blue shading indicates homologous genes between species”. Blue is grey.
- Figure legend Figure S4c: “Purple triangle indicates position of S.oct-cen” Is not S.cry-cen? “Homologies lies within cTART14 elements (also present at S. cryophilus cen1 and cen3)”. Should be “cen1 and cen2”

We thank the reviewer for pointing out these errors which we have now corrected.

Reviewer #3 (Remarks to the Author):

This manuscript reports on the structure and organization of centromeres in related fission yeasts *S. pombe*, *S. octosporus*, and *S. cryophilus* (and *S. japonicus*). The authors use long read DNA sequencing to fully assemble each genome and repetitive centromeric sequences and combine this with ChIP-seq CENP-A and H3K9me2 localization, and functional analysis. Their results reveal the conserved and diverged features of centromere organization including highly conserved syntenic 5S RNA and tRNA genes across centromeres with the latter appearing to play a conserved boundary role. The authors also conclude that non-homologous central core sequences from *S. Octosporus* and *S. pombe* play conserved roles in de novo centromere establishment. These results are very interesting and of broad general interest. Although the results are mostly descriptive, I really enjoyed reading this paper and have only one concern noted below regarding the second conclusion.

We thank the reviewer for her/his positive comments.

I think the authors are overstating the case for cross-species conservation of cen DNA features that promote CENP-A assembly. Their data only show that *S. octosporus* cnt DNA is permissive for CENP-A assembly when placed next to *S. pombe* outer cen repeat sequences that can promote CENP-A assembly and centromere function. The underlying sequence has not been demonstrated to have *de novo* centromere assembly function.

We have shown that when placed in a similar context to *S. pombe* central core (ie next to *S. pombe* outer repeat) *S. octosporus* (and now *S. cryophilus* – see below) central domain DNA permits *de novo* centromere assembly function. We agree that we have not shown that *S. octosporus*-only centromere DNA can establish CENP-A chromatin, but the purpose of the *S. pombe* outer repeat is only to provide heterochromatin which we already know is required for establishment of CENP-A chromatin (Folco et al, 2008) and to provide robust sister-centromere cohesion and thus reasonable segregation function (Bernard et al, 2001). The use of *S. pombe* outer repeat allows specific investigation of the ability of *S. oct* and *S. cry* central core sequences to establish CENP-A chromatin and centromere function, without the additional need for heterochromatin formation on non-homologous flanking repeats from these other species. For example, if we used an *S. octosporus* outer repeat and a central domain and no centromere establishment was observed it could be due to defective heterochromatin formation (required for both CENP-A establishment and centromeric cohesion on minichromosomes).

We have now performed similar cross-species analyses for *S. cryophilus* centromere sequences as were performed for *S. octosporus* centromere sequences in the original manuscript. For *S. cryophilus* we inserted regions from *Sc-cnt1* (homologous to *Sc-cnt2*) and *Sc-cnt3* into a plasmid containing *S. pombe* outer repeat. The resulting minichromosome plasmids were able to establish centromere segregation function and assemble CENP-A chromatin in *S. pombe* cells (albeit with a lower establishment frequency than *S. pombe* and *S. octosporus* sequences). This new data is presented in Supplementary Figure S8 and mentioned in the text. Limited analyses were also performed using *S. japonicus* Tj7 retrotransposon DNA that is associated with high levels of CENP-A in *S. japonicus*. However, minichromosomes bearing Tj7 did not convincingly establish centromere function or CENP-A chromatin in *S. pombe* (Supplementary Fig. 6).

The authors previous work has shown that other non-related sequences can also be permissive for CENP-A assembly in this assay. The authors should tone down this point in both their abstract and at the end of the text.

It is unclear to which of our previous studies the reviewer is referring. To our knowledge, we (or others) have not previously shown that other non-centromeric sequence can assemble CENP-A/functional kinetochores in the minichromosome assay presented in Fig. 6. We have previously shown that when CENP-A is overexpressed it can accumulate at particular promoters, or other locations, such as gene bodies, when FACT or Spt6 is mutated to allow more promiscuous CENP-A deposition (Choi et al, 2012). However, in these cases, CENP-A assembles at very low levels compared to the levels we see on assemble on central core sequences - central core is a preferred substrate (Catania et al, 2015). In addition, others have shown that following deletion of an endogenous centromere, CENP-A can assemble near telomeres to form neocentromeres at low frequencies on recovered chromosomes (Ishii et al, 2008). Overexpressed CENP-A can accumulate at low levels in these locations to some extent (Castillo et al, 2007). Supported by our k-mer PCA (Figure 4), we speculate that these neocentromere regions share some features in common with central domain DNA of *S. pombe*, *S. octosporus* and *S. cryophilus*. We have also shown that CENP-A can spread onto marker genes inserted in the central core, but this is also in the context of flanking functional central core sequences and not in an establishment assay (Castillo et al, 2007). It is therefore not clear to us what point the reviewer wants us to make less forcefully, however we have adjusted the abstract.

The conservation of tDNA and 5S gene arrangements described by the authors are particularly interesting. This paper will become a valuable resource to the community and a starting point for further analysis of these and other centromere feature revealed by the results.

We are pleased that the referee recognises the usefulness of our detailed annotation of

Schizosaccharomyces centromeres and genomes to the community.

Reviewer #4 (Remarks to the Author):

This is a very well written manuscript that was a real pleasure to review. In brief, it describes the assembly of three *Schizosaccharomyces* genomes with a focus on the repetitive genome components specifically of the centromeres and telomeres. The quality of the figures is similarly exceptional. In short, it is a great paper and my comments are minor in scope and intended to clarify a few points for the non-expert reader.

We thank the reviewer for these compliments.

Key claims of the manuscript include 1) conservation, over millions of years, of syntenic clusters of tRNA and rRNA genes and 2) a non-homologous centromere core sequence from one species providing centromere function in another. The claims are well substantiated with the caveats listed below, which might justify minor changes or additions to the discussion.

The syntenic arrangement of tRNA genes near *S.cry-cen1* and 2 and *S.oct-cen2* and 3 together with the CENP-A and H3K9me2 ChIP data does suggest that they constitute a barrier to heterochromatin spread as suggested by the authors. However, the authors do not comment on the two CENs that are not flanked by tRNAs are the only ones composed of inverted repeats (*S.cry-cen3* and *S.oct-cen1*). LTRs are indicated for *S.cry-cen3* as fulfilling this “demarcation” function, but no such LTRs are marked in the apparently paralogous *S.oct-cen1* (Fig. 2b/c). Thus the inverted nature of both of these CENs seems to me to be at least as important as the presence of flanking LTRs in one of them.

In fact, all central cores in *S. pombe*, *S. octosporus* and *S. cryophilus* are flanked by inverted repeats. But those flanking *S.cry-cen3* and *S.oct-cen1* are indeed very long compared to the others. We suggest that the LTRs at *S.cry-cen3* are candidates for boundary elements due to the striking precise coincidence of the LTRs with the transition from CENP-A to heterochromatin (this will require rigorous testing). Also, as mentioned in the text, LTRs, like tRNA genes, are nucleosome-depleted regions which is a known feature of boundary elements. Because no full-length retrotransposons (only remnants) are present in the *S. octosporus* genome we do not know what intact *S. octosporus* LTRs look like and, therefore, if they exist within the inverted repeats of *S.oct-cen1*. However, we now note in the text that the transition between CENP-A and heterochromatin is far less sharply demarcated at *S.oct-cen1* than for any of the other centromeres, so they may well not have a strong boundary/demarcation element, and perhaps an unfocused/ imprecise transition zone exists that coincides with the long, inverted repeats at *S.oct-cen1*. We have added a sentence to highlight this possibility.

The experiments describing the functioning of the So CNT in *S. pombe* are fascinating and very nicely done. In light of the K-mer analysis, it would be nice to know how critical the CNT repeats are, i.e. can any DNA consisting of “CENP-A-favored” kmers and flanked by the outer repeats function as centromere?

In the future we intend to perform further dissection of the composition and functional requirements with respect to the various central domain sequences. Such experiments will require manipulation of many parameters and the generation of many versions of the central domain template. Such detailed experiments are complex and therefore beyond the scope of the current study.

Do previous publications exist where such swaps have been attempted?

To our knowledge no other studies have been performed in establishment assays where the ability of central domain DNA from one fission yeast species to establish CENP-A chromatin and a functional centromere has been tested in another species. In mammalian cells mouse minor satellite DNA can establish functional centromeres in certain circumstances in human cells (Okada et al, 2007) – this is mentioned in the text.

Slightly more discussion of why the “NA” and “ND” clones did not work (see comment below regarding Figure 6d) could be enlightening.

NA means 'not applicable'. Because these plasmids did not establish centromere function, they could not be assessed in a minichromosome loss assay for chromosome segregation competence. This is now noted in the Figure 6 legend. ND means not determined. As numerous other plasmids were quantified for minichromosome loss we have elected to remove this one minichromosome from the table as quantification of its loss rate would add little to the manuscript.

A little more background regarding these experiments might also be useful, e.g.: Why was only one *S. pombe* outer repeat used to test So-cnt in *S. pombe*, when the actula *S. pombe* centromeres are flanked by inverted repeats? Is this a technical limitation of cloning inverted repeats in *E. coli* or *S. cerevisiae*?

Plasmids with long inverted repeats are indeed highly unstable in *E. coli*. Therefore the standard configuration of minichromosome plasmids that have been used to study establishment function in *S. pombe* for many years is the use of repeats on only one side of a central domain (e.g. Hahnenberger & Clarke, 1991; Baum et al, 1994; Hahnenberger et al, 1989; Folco et al, 2008; Catania et al, 2015). A sentence and reference have been added to the Methods explaining the use of 'one-sided' minichromosomes.

What do the circles above the "K-rpt" and "S.oct-cnt" in Figure 6a indicate (nucleosome positions)?

The circles were intended to be a cartoon depiction of some nucleosomes. To avoid confusion, they have been removed and replaced by the words 'heterochromatin' and 'CENP-A chromatin?'.

And is the "K-rpt" the same sequence that is orange in Fig. S5A? IF not, it might be worth highlighting that particular repeat in Fig. S5A.

We have changed the colouring in Supplementary Fig 5 so that the *dg* repeat is dark orange and the *dh* repeat is tangerine, and modified the legend accordingly. The *dg/dh* nomenclature was adopted by Mitsuhiro Yanagida and colleagues, whilst the *K/K'/K"/B/L* nomenclature was used by Louise Clarke and colleagues. The *dg* repeat approximates to a *K* repeat (which differ in orientation/organisation at different centromeres). We have marked the outer-repeat regions present in minichromosome plasmids on Fig S5 and added additional information to the legend.

Minor comments:

The authors may need to define the term "tDNA", which I suspect is shorthand for tRNA genes?

We have changed all instances of tDNA to tRNA gene.

Page 5: "benefitting chromosome segregation selected" – comma before "selected"

Agreed.

Page 5: "TFIIIC" – may need definition, though easily googled

This is a standard abbreviation for a well-known transcription factor. To define it in the text will needlessly complicate the sentence.

Page 15: Figure 4a legend needs a period

Corrected.

Figure 6d: "ND" and "NA" are not defined in figure or legend. It is critical to know what these mean in order to evaluate the results.

This information has been added to the figure legend for Fig 6.

Figure S1 legend: "Blue shading" should be "Grey shading"?

This has been changed in the text.

Figure S3A legend: Mention that black circles = centromeres, as not listed in Figure.

This has been changed in the text.

REVIEWERS' COMMENTS:

Reviewer #1 (Remarks to the Author):

This manuscript provides an in-depth study of the conservation of centromere sequences, comparing four *Schizosaccharomyces* sequences. Although sequence was not conserved, the domain organization was conserved. In addition, introduction of centromere core regions from *S. octosporus* and *S. cryophilus* were able to assemble CENP-A chromatin and assemble kinetochores. Their results suggested that these regions, although lacking sequence conservation, had elements that allowed centromere assembly.

I am pleased with the revision, as it addressed the main issues that arose in my review. Additional analysis of *S. cryophilus* showed that the introduction of the sequences was also able to assemble centromeric DNA. The addition of *S. japonicus* in the assay showed that its centromere sequence likely does not share the same properties as *S. octosporus* and *S. cryophilus* in assembling centromeric DNA in *S. pombe*.

Overall, I think this wealth of sequencing data will be incredibly useful to many groups of researchers, including those who study centromeres, those who study sequence evolution, and those who use *Schizosaccharomyces* as model systems. In addition, the main findings of the paper are intriguing and impactful to our understanding of how centromeres are specified.

Reviewer #2 (Remarks to the Author):

In this revised version authors have successfully addressed my questions
Dr. F. Azorin

Reviewer #3 (Remarks to the Author):

The authors have addressed my concern and I enthusiastically support publication.

Reviewer #4 (Remarks to the Author):

The edits of (and additions to) the manuscript have clarified all of the issues I had raised earlier. I wish that an experimental test of the k-mer centromere functions could have been included in this paper, and am certainly looking forward to that next publication. As I stated before, this is a beautiful paper that adds significant data and analysis to a field that is of interest to many scientists, even in the absence of experimental k-mer data.

Response to Reviewers

We thank the reviewers for their time, valuable input and support of publication.

Reviewer #1 (Remarks to the Author):

This manuscript provides an in-depth study of the conservation of centromere sequences, comparing four *Schizosaccharomyces* sequences. Although sequence was not conserved, the domain organization was conserved. In addition, introduction of centromere core regions from *S. octosporus* and *S. cryophilus* were able to assemble CENP-A chromatin and assemble kinetochores. Their results suggested that these regions, although lacking sequence conservation, had elements that allowed centromere assembly.

I am pleased with the revision, as it addressed the main issues that arose in my review. Additional analysis of *S. cryophilus* showed that the introduction of the sequences was also able to assemble centromeric DNA. The addition of *S. japonicus* in the assay showed that its centromere sequence likely does not share the same properties as *S. octosporus* and *S. cryophilus* in assembling centromeric DNA in *S. pombe*.

Overall, I think this wealth of sequencing data will be incredibly useful to many groups of researchers, including those who study centromeres, those who study sequence evolution, and those who use *Schizosaccharomyces* as model systems. In addition, the main findings of the paper are intriguing and impactful to our understanding of how centromeres are specified.

Reviewer #2 (Remarks to the Author):

In this revised version authors have successfully addressed my questions
Dr. F. Azorin

Reviewer #3 (Remarks to the Author):

The authors have addressed my concern and I enthusiastically support publication.

Reviewer #4 (Remarks to the Author):

The edits of (and additions to) the manuscript have clarified all of the issues I had raised earlier. I wish that an experimental test of the k-mer centromere functions could have been included in this paper, and am certainly looking forward to that next publication. As I stated before, this is a beautiful paper that adds significant data and analysis to a field that is of interest to many scientists, even in the absence of experimental k-mer data.